# EcoMapper: Generative Modeling for Climate-Aware Satellite Imagery

**Muhammed Goktepe** [* 1]  **Amir Hossein Shamseddin** [* 1]  **Erencan Uysal** [1]  **Javier Muinelo Monteagudo** [1]
**Lukas Drees** [2]  **Aysim Toker** [3]  **Senthold Asseng** [4]  **Malte von Bloh** [4]

## Abstract

Satellite imagery is essential for Earth observation, enabling applications like crop yield prediction, environmental monitoring, and climate change assessment. However, integrating satellite imagery with climate data remains a challenge, limiting its utility for forecasting and scenario analysis. We introduce a novel dataset of 2.9 million Sentinel-2 images spanning 15 land cover types with corresponding climate records, forming the foundation for two satellite image generation approaches using fine-tuned Stable Diffusion 3 models. The first is a text-to-image generation model that uses textual prompts with climate and land cover details to produce realistic synthetic imagery for specific regions. The second leverages ControlNet for multi-conditional image generation, preserving spatial structures while mapping climate data or generating time-series to simulate landscape evolution. By combining synthetic image generation with climate and land cover data, our work advances generative modeling in remote sensing, offering realistic inputs for environmental forecasting and new possibilities for climate adaptation and geospatial analysis.

## 1. Introduction

Satellite imagery is a critical resource for a variety of research and industrial applications, ranging from environmental monitoring and climate research (de Araújo et al., 2025; Hussain et al., 2022) to agriculture (Khanal et al., 2020), archaeology (Marín-Buzón et al., 2021), resource exploration (Shirmard et al., 2022), and human development mapping (Burke et al., 2021b). It provides valuable spatial and temporal data, often serving as the primary or supplementary source for prediction and analysis models, including crop yield forecasting (Khaki et al., 2021; Muruganantham et al., 2022; von Bloh et al., 2023), forestry surveillance (Fassnacht et al., 2024), and disaster management (Burke et al., 2021a). In many regions of the world, satellite data is the only feasible means of acquiring near real-time information about environmental conditions.

However, the use of remote sensing is hampered by significant challenges, including cloud coverage, atmospheric distortions, and temporal resolution constraints (Dubovik et al., 2021). Cloud coverage, in particular, renders satellite imagery unusable for large parts of the world, disrupting satellite observations in cloud-prone areas for days or even weeks, depending on the season (King et al., 2013; Mercury et al., 2012). These operational challenges not only hinder real-time monitoring but also raise a critical conceptual gap: the need to integrate satellite imagery with future climate scenarios to enhance prediction accuracy. While various datasets support machine learning applications to address these challenges, most are task-specific or regionally constrained, limiting their generalizability (Clasen et al., 2024; Christie et al., 2018; Schneider et al., 2023; Van Etten et al., 2018). To address this, we introduce a comprehensive remote sensing dataset—one of the largest to date—featuring over 2.9 million Sentinel-2 RGB images linked with climate data, enabling more robust and scalable applications across diverse environmental conditions. Recent advances in multimodal foundation models for remote sensing have significantly improved generation of synthetic satellite imagery across domains (Hong et al., 2024; Khanna et al., 2024; Liu et al., 2024a; Yu et al., 2024b). But a key gap remains in generative models capable of producing location-specific satellite imagery conditioned on future climatic conditions. This limitation hinders predictive applications such as seasonal crop yield forecasting and the assessment of climate change impacts on land cover (Iizumi et al., 2018; Zachow et al., 2024). These models rely on detailed weather and climate projections to improve their accuracy, but often

---

[*]Equal contribution [1]Technical University of Munich, School of Computation, Information and Technology, Germany [2]University of Zurich, Department of Mathematical Modeling and Machine Learning, EcoVision Lab, Switzerland [3]Technical University of Munich, School of Computation, Information and Technology, Dynamic Vision and Learning Group, Germany [4]Technical University of Munich, School of Life Sciences, Department of Life Science Engineering, HEF World Agricultural Systems Center, Chair of Digital Agriculture, Germany. Correspondence to: Malte von Bloh <malte.von.bloh@tum.de>.

*Proceedings of the 42$^{nd}$ International Conference on Machine Learning*, Vancouver, Canada. PMLR 267, 2025. Copyright 2025 by the author(s).

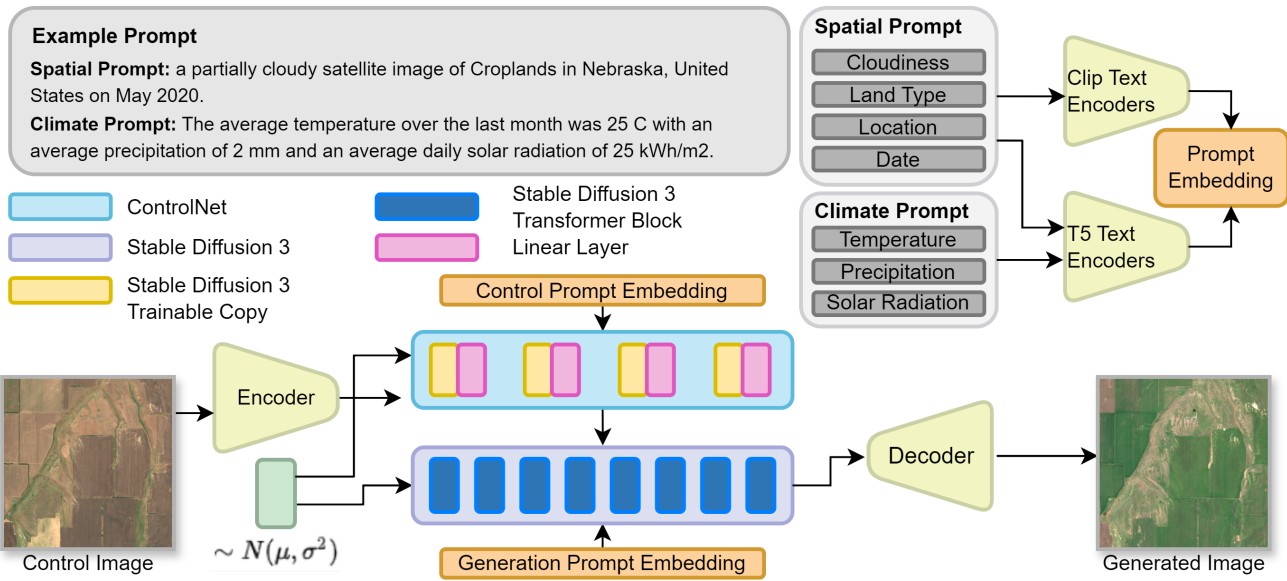

*Figure 1.* Our framework enables multi-conditional (text+image) satellite image generation using Stable Diffusion 3 (Esser et al., 2024) and ControlNet (Zhang et al., 2023). On the text-input side, the model takes detailed spatial and climatic prompt embeddings created by CLIP and T5 text encoders. On the image-input side, ControlNet is fine-tuned to process an image for spatial guidance. Both is passed to pre-trained Stable-Diffusion 3, which generates a location- and climate specific satellite image, aligned with actual spatial characteristics.

depend on historical or approximated satellite data in the absence of integration with climate scenarios. This results in a qualitative gap between predicted environmental conditions and available imagery, complicating the representation of dynamic changes in land cover or vegetation states, reducing the predictive power of these models (Ebel et al., 2020; Jozdani et al., 2022). These challenges underscore the need for synthetic satellite imagery to enhance datasets and provide realistic projections for future conditions. In this paper, we introduce a novel approach to generate satellite images conditioned on geographic-climate prompts using Stable Diffusion 3. Our method enables the simulation of how weather and climate affect Earth's surface - generating synthetic images that can support forecasting models (e.g., crop yield prediction or land cover classification), visualize climate change models under various scenario assumptions, and fill observational gaps in regions affected by persistent cloud cover. The approach is globally applicable and generates realistic images with 10-meter spatial resolution across diverse vegetation types (e.g., cropland, broadleaf forests, savannas), using information about location, land cover type, and climate conditions. We propose two innovations:

1. A text-to-image generation model that leverages Stable Diffusion 3 with climate prompt engineering.
2. A multi-conditional (text+image) framework utilizing ControlNet, which preserves spatial features and enables the generation of time series.

Fig. 1 illustrates the concept. To support this research, we curated a dataset of over 2.9 million RGB satellite images from 104,424 locations worldwide, sourced from Sentinel-2 (Drusch et al., 2012). This dataset spans the whole Earth categorized in 15 vegetation zones and eight years of historical data. Together, these contributions advance the application of generative models in remote sensing and offer novel solutions for a variety of environmental monitoring challenges.

## 2. Related Work

### 2.1. Diffusion Models

Diffusion models are a powerful class of generative models, achieving state-of-the-art performance in high-quality image synthesis across diverse domains (Ho et al., 2020; Khanna et al., 2024; Zhang et al., 2023). They operate by iteratively denoising random Gaussian noise through a learned reverse-diffusion process, producing realistic samples. These models excel in tasks like image super-resolution, inpainting, and domain adaptation (Manvi et al., 2024; Toker et al., 2024), and have been effectively applied to satellite imagery synthesis by leveraging control images and textual prompts to maintain spatial structure and stylistic fidelity (Sastry et al., 2024). Foundational work by Sohl-Dickstein et al. (2015) introduced diffusion probabilistic models, later refined by Ho et al. (2020) with a simplified denoising objective. Score-based generative modeling (Song et al., 2021) and classifier-free guidance (Dhariwal & Nichol, 2021) have enhanced their adaptability and output quality.

## 2.2. Satellite Image Generation

In remote sensing, image generation has also been enhanced by diffusion models, with models like Stable Diffusion (SD) fine-tuned to generate satellite images from textual descriptions (Liu et al., 2024b). Khanna et al. (2024) enriched in their "DiffusionSat" the SD model's input by incorporating geo-location and sampling time as prompts, enabling the generation of high-quality satellite images tailored to specific geographic and temporal conditions. Their experiments focused on generating diverse land cover types, including urban areas, croplands, and forests, demonstrating the model's ability to capture fine-grained details such as building structures and vegetation patterns.

Diffusion models have also been widely adopted for image-to-image generation, producing realistic satellite images from guiding inputs such as maps, semantic layouts, and multi-modal data. Espinosa & Crowley (2023) successfully generated satellite images conditioned on historical maps, focusing on urban and rural landscapes to simulate historical and future land use changes. Tang et al. (2024) refined the generation process by integrating both global (e.g., textual descriptions) and local (e.g., depth maps, segmentation masks) control information, expanding the scope of satellite image synthesis.

Satellite image datasets range from small task-specific collections to large general-purpose sets, with unlabeled remote sensing data often exceeding one million images, while labeled datasets are typically smaller in size. Reben (Clasen et al., 2024) and EarthView (Velazquez et al., 2025) support general-purpose and self-supervised learning, with Earth-View comprising over 15 terapixels of multi-source imagery. fMoW (Christie et al., 2018) and EuroCrops (Schneider et al., 2023) provide labeled data, targeting functional land use and harmonized crop types. Multimodal and cloud-robust datasets such as SEN12MS-CR-TS (Ebel et al., 2022) and DiffCR (Zou et al., 2024) offer paired radar-optical time series for cloud removal. In contrast, MetaEarth (Yu et al., 2024b) presents a generative foundation model trained on multi-resolution imagery for large-scale image synthesis. In contrast, climate-integrated datasets have so far been smaller and highly specialized for research applications (Nath et al., 2024; Requena-Mesa et al., 2021).

## 2.3. Integrating Climate Data

When processing datasets with high-dimensional climate variables like cyclone distribution, cloud cover, and water vapor, diffusion models have proven effective (Liu et al., 2024b). Hatanaka et al. (2023) used cascaded diffusion models to generate high-resolution cloud cover images, while Nath et al. (2024) employed multi-stage diffusion frameworks for precipitation and cyclone forecasting. Gao et al. (2023) and Leinonen et al. (2023) unified precipitation now-casting within single diffusion models, achieving state-of-the-art results by capturing complex spatiotemporal relationships. DiffCast from Yu et al. (2024a) outperforms previous works in the Critical Success Index (CSI) - which measures the fraction of correctly predicted precipitation events relative to all observed or predicted events - by 15.59 %.

## 3. Preliminaries

### 3.1. Diffusion

Diffusion models aim to produce images by reversing a stochastic Gaussian noising process. Given an input image, a noisy input $x_t$ is created by adding Gaussian noise:

$$x_t = \sqrt{1 - \beta_t} \cdot x_{t-1} + \sqrt{\beta_t} \cdot \epsilon, \quad \epsilon \sim \mathcal{N}(0, I). \quad (1)$$

The parameters $\beta_t$ denote the noise variance at each timestep $t$, the higher the $t$ the more the added noise. The aim of the diffusion model is to denoise a random sample using a neural network to predict its noise vectors, effectively learning the mapping $x_t \mapsto x_{t-1}$. This results in a stochastic generative model $M$ that maps from a predefined noise distribution $\mathcal{N}(0, I)$ to generated images.

$$M(z) = x_0, \quad z \sim \mathcal{N}(0, I). \quad (2)$$

Latent diffusion models are a type of generative model that applies the diffusion process within a lower-dimensional latent space, which is obtained by encoding the input images. This reduces computational cost compared to traditional diffusion models. A decoder then reconstructs the image from the latent space. Conditional architectures, including Stable Diffusion 2, guide the denoising process by incorporating an additional signal $C(\cdot)$, such as text or images. This enables the generation of meaningful outputs that align with the control signal and the specific task for which the model was trained.

$$M\big(z, C(z)\big) = x_0, \quad z \sim \mathcal{N}(0, I). \quad (3)$$

### 3.2. ControlNet

ControlNet (Zhang et al., 2023) enhances diffusion models by integrating explicit spatial control into the generative process. Let $F(\mathbf{x})$ denote a neural network block from the original architecture, which has frozen parameters internally (omitted here for brevity). The new control block modifies its output by adding

$$\mathbf{y} = F(\mathbf{x}) + Z\Big(F\big(\mathbf{x} + Z(\mathbf{c}; \boldsymbol{\theta}_1); \boldsymbol{\theta}_{\text{copy}}\big); \boldsymbol{\theta}_2\Big), \quad (4)$$

where $\boldsymbol{\theta}_{\text{copy}}$ is a trainable copy of the original parameters, and $\boldsymbol{\theta}_1$ and $\boldsymbol{\theta}_2$ are the parameters of zero modules (e.g. convolution layers initialized to zero). This design ensures that the control block has minimal initial effect on the main block, functioning similarly to a skip connection.

# 4. Material and Methods

## 4.1. The EcoMapper Dataset

We present the EcoMapper dataset that combines Sentinel-2 RGB images with corresponding metadata, assembled using Google Earth Engine, Sentinel Hub and NASA Power (Drusch et al., 2012; Milcinski et al., 2019; NASA, 2025; Phan et al., 2020).

### 4.1.1. SATELLITE IMAGERY

The dataset includes 104,424 unique geographic locations, randomly sampled from 15 distinct land cover classes (Phan et al., 2020) excluding water bodies as shown in Figure 2. For each location, we selected one monthly observation for a two year period based on the least cloudy day, resulting in a sequence of 24 images per location. The two years of observation are randomly distributed between 2017 and 2022. The test set consists of 5,500 unique geographic locations, each monitored monthly over a 96-month period from 2017 to 2024. This ensures sufficient spatial and temporal independence in the evaluation, enabling robust assessment of the model's generalization across diverse regions and unseen climate conditions. With a spatial coverage of $\sim 26.21$ km$^2$ per observation the overall dataset covers $\sim 2,704,000$ km$^2$, accounting for $\sim 2.05$ % of Earth's terrestrial area. An excerpt of the dataset is published in the Github repository, the full dataset is available at the universities servers. For more dataset details we refer to A.1.

### 4.1.2. CLIMATE DATA

Each sampled location is enriched with metadata, including geographic location (longitude and latitude), observation date (month and year), land cover type, and cloud coverage (in %). We incorporated average monthly temperature, solar radiation, and total precipitation from NASA Power (NASA, 2025), as these factors mainly drive vegetation growth, energy availability and water balance, which in turn influence agricultural conditions, forestry, biodiversity, and land cover (Pielke Sr et al., 2011).

## 4.2. Generative Models

Our goal is to synthesize satellite imagery conditioned on geographic and climate metadata, enabling realistic projections of environmental conditions. To achieve this, we leverage state-of-the-art generative models for two key tasks: text-to-image generation and multi-conditional image generation.

For text-to-image generation, we employ generation models that synthesize satellite images based on structured metadata prompts. Additionally, we introduce a multi-conditional generation approach using a ControlNet-enhanced model,

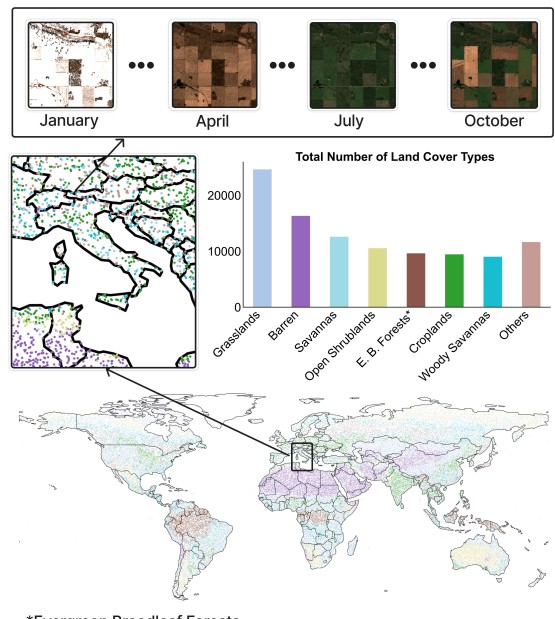

*Figure 2.* The 104,424 locations were sampled globally across 15 land cover types, providing a representative distribution of Earth's land surface. Grasslands and sparsely vegetated regions dominate, followed by forested areas and croplands, with additional categories summarized into "Others" (mixed-, evergreen needleleaf forest, permanent wetlands, cropland/natural mosaics, urban, closed shrubland and decidiuous needleleaf forest). Each location includes a time series of 24 months (training) or 96 months (testing).

which preserves the spatial structure of an input image while mapping climate-induced variations onto it. By leveraging our dataset, we demonstrate how environmental changes can be visually represented by modifying climate metadata in the generation process.

We evaluate two generative models for their ability to integrate climate metadata into satellite image synthesis:

Stable Diffusion 3 (SD3) from Esser et al. (2024) - A multimodal latent diffusion model incorporating CLIP and T5 text encoders for flexible prompt conditioning. We fine-tune SD3 using our dataset to generate realistic satellite imagery conditioned on geographic, climatic, and temporal metadata. A key challenge is the representation of continuous climate variables, which we address through structured prompt engineering. DiffusionSat from Khanna et al. (2024) - a specialized foundation model for satellite imagery, extending Stable Diffusion 2 with a dedicated metadata embedding layer for numerical conditioning. This architecture encodes key spatial and temporal attributes, including lati-

tude, longitude, timestamp, ground sampling distance, and cloud cover. Unlike generic diffusion models, DiffusionSat is explicitly designed for remote sensing tasks, including super-resolution, inpainting, and temporal prediction.

### 4.2.1. TEXT-TO-IMAGE GENERATION

We compare multiple configurations of Stable Diffusion 3 and DiffusionSat, with and without fine-tuning, to assess their capacity for climate-aware satellite image synthesis. The base SD3 model, leveraging a T5 encoder, allows full climate prompt conditioning. The base DiffusionSat model, limited to one text encoder with 77 tokens, was only conditioned on month, year, cloudiness, and land cover type due to its predefined metadata embedding structure. To enable DiffusionSat fine-tuning with additional climate metadata, we modified its SatUNet architecture by introducing 10 metadata layers. We initialized this adapted network with pretrained DiffusionSat weights while randomly initializing the new layers, followed by full-model fine-tuning. For a fair comparison, both models were trained at 512×512 resolution, which aligns with DiffusionSat's original training setup. Additionally, SD3, which supports higher resolutions, was tested in a fine-tuned experiment at 1024×1024 resolution. In summary we evaluate:

1. Baseline models: Both models were evaluated without fine-tuning at 512×512 resolution.
2. Fine-tuned models: Both models were fine-tuned (-FT) with climate metadata at 512×512 resolution.
3. High-resolution SD3: SD3-FT-HR was fine-tuned with climate metadata at 1024×1024 resolution.

### 4.2.2. CLIMATE-AWARE SENSITIVITY ANALYSIS

To assess the sensitivity of the SD3-FT-HR model to climate variables and ensure that performance gains stem from meaningful climate conditioning rather than spurious correlations with month, land cover, or location we perform a targeted sensitivity analysis. We evaluate the model's ability to incorporate climate effects into satellite image generation by testing it under extreme conditions, ranging from dry boreal (cold, dry) to humid tropical (hot, wet) climates. These variations span multiple land cover types and regions, enabling us to determine whether the model captures genuine climate influences or merely exploits dataset correlations.

### 4.2.3. MULTI-CONDITIONAL IMAGE GENERATION

For the task of multi-conditional (text+image) image generation, we utilize a fine-tuned Stable Diffusion 3 model enhanced with LoRA (Low-Rank Adaptation). This model, trained at a 512×512 resolution, serves as a foundational prior for generating high-quality and contextually relevant outputs. To condition the image generation process on both

spatial structure and climate dynamics, we incorporate a dual-conditioning mechanism using ControlNet. ControlNet extends Stable Diffusion by introducing trainable neural layers that guide the denoising process using an external control signal. In our approach, two critical conditioning signals are used: Satellite imagery from previous months as a control signal that preserves the spatial structure of the generated image, ensuring that landforms, urban layouts, and other geographical features remain intact. This also enables the model to incorporate temporal changes over time, reflecting real-world environmental shifts. Climate Prompts: A textual conditioning mechanism that specifies climatic and atmospheric conditions under which the satellite image should be generated. By combining these two conditioning factors, the model is capable of generating realistic satellite images that integrate climate variations while maintaining spatial consistency. This method supports time-series generation, allowing the simulation of landscape evolution under changing climate conditions.

## 4.3. Prompt Structure

We design two types of prompts to effectively condition satellite image generation: a spatial prompt, which encodes metadata, and a climate prompt extending it with environmental details. Both prompts leverage the text encoders of Stable Diffusion 3, with spatial information processed by CLIP and climate data handled by the T5 encoder.

1. **Spatial Prompt**: Captures fundamental metadata, including land cover type, location, date, and cloudiness. This ensures that the generated images align with the geographic and temporal context.
2. **Climate Prompt**: Extends the spatial prompt by incorporating monthly climate variables temperature, precipitation, and solar radiation providing additional environmental conditioning for image generation.

The structured prompt follows the format: "A satellite image of [land cover type] in [location] on [date]. The average temperature was [temperature], with [precipitation] and [solar radiation]." This format ensures the generated images remain contextually and environmentally accurate by integrating both spatial and climatic factors. For example: *"A satellite image of croplands in Northern Cape, South Africa, on October 2019. The average temperature over the last month was 20°C, with an average precipitation of 0 mm and an average daily solar radiation of 25 W/m²."* This structured prompting framework enables effective conditioning across spatial and environmental dimensions. Additional details on the prompting strategy and comparative results for different prompt formulations are provided in Appendix A.3.5.

### 4.3.1. METRICS

To evaluate the quality, diversity, and perceptual fidelity of the generated satellite images, we use five established metrics: FID, LPIPS, SSIM, PSNR, and CLIP Score. FID and LPIPS assess realism and perceptual similarity, while SSIM and PSNR measure structural consistency and reconstruction quality. CLIP Score evaluates text-image alignment. A detailed description is provided in A.2.

## 5. Results

### 5.1. Text-to-Image Generation

We test our models on 5,500 worldwide locations across all land cover types, using eight years of satellite data with monthly observations. As shown in Tab. 1, the baseline models from SD3 and DiffusionSat had the lowest evaluation scores. However, DiffusionSat demonstrated superior performance over SD3, showing advantages from the satellite-specific pretraining. Fine-tuning significantly improved both models across all metrics: SD3-FT achieved higher CLIP, SSIM, and PSNR scores, while DiffusionSat-FT excelled in FID and LPIPS. The best-performing model in terms of FID was SD3-FT-HR, which produced the highest-resolution images.

The qualitative results in Fig. 3 demonstrate the capability of our models in generating realistic satellite images conditioned on geographic and land type metadata. In particular, the models excel at capturing the structured patterns of croplands/grasslands, such as those seen in Kazakhstan and Paraguay, where regular field patterns are faithfully reproduced. All three models effectively preserve the essential structure of these landscapes. In mountainous regions like Canada, the models successfully capture the distinct features of snow-covered terrains and rocky surfaces. While both architectures handle snow coverage well, SD3-FT-HR excels at preserving the sharp contrasts between snow and rock formations, providing finer detail compared to DiffusionSat in this context. In grassland regions of Paraguay, the models represent the expansive, flat terrain with sparse vegetation effectively, capturing the homogeneous structure typical of grasslands. All models manage to represent the broadness of these regions, though SD3 models show a slight improvement in capturing the subtle variations in vegetation density. For wetland areas DiffusionSat captures the water presence with high fidelity, while SD3 also effectively represents the wetland's dynamic structure, with both models excelling in maintaining the spatial features, though with slight differences in texture and detail depending on the model configuration. A comparison of both pretrained and fine-tuned models with details about the climate data can be found in Appendix 10.

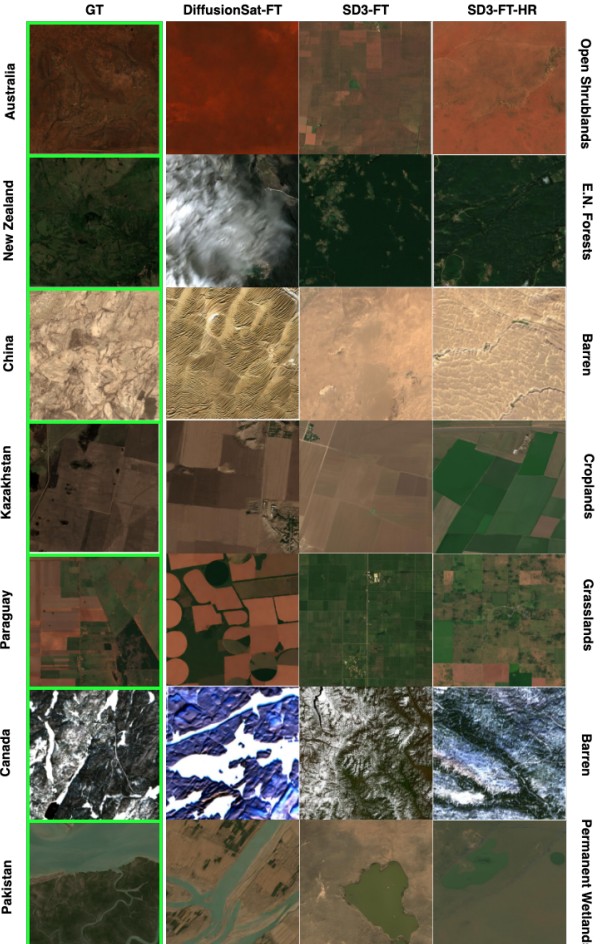

*Figure 3.* Qualitative comparison of satellite images generated by different models. The columns represent Ground Truth (GT), DiffusionSat-FT, SD3-FT, and SD3-FT-HR, where FT denotes fine-tuned and HR high-resolution models. Each row specifies the country on the left and the corresponding land cover type on the right.

### 5.2. Climate-Aware Sensitivity Analysis

The sensitivity experiments in Fig. 4 show a notable variation in generated vegetation across different climate conditions. In humid environments, the generated images exhibit denser vegetation, whereas in cold and dry conditions, the outputs display barren or snow-covered landscapes. Across most land cover types, there is an observable increase in simulated vegetation as climate inputs shift towards warmer and more humid conditions. Conversely, drier and colder conditions result in sparser vegetation and more exposed land surfaces. However, the magnitude of these effects varies by region. In areas where the prompted climate conditions deviate strongly from the typical ones of the region, the model exhibits a weaker response or no observable change.

*Table 1.* Quantitative comparison of text-to-image generation models. The base models refer to the original models without fine-tuning, while the "-FT" models have been fine-tuned at a resolution of 512×512 pixels. The SD3-FT-HR model was fine-tuned and trained at a higher resolution of 1024×1024 pixels.

| Model | FID ↓ | CLIP ↑ | SSIM ↑ | PSNR ↑ | LPIPS ↓ |
|---|---|---|---|---|---|
| **SD3** | 157.36 | 0.29 | 0.14 | 8.25 | 0.85 |
| **DiffusionSat** | 115.00 | 0.31 | 0.25 | 10.18 | 0.80 |
| **SD3-FT** | 77.9 | **0.33** | **0.39** | **14.24** | 0.70 |
| **DiffusionSat-FT** | 59.8 | 0.30 | 0.33 | 12.14 | **0.68** |
| **SD3-FT-HR** | **49.48** | 0.31 | 0.36 | 12.01 | 0.69 |

*Table 2.* SD3-FT Performance on extreme weather conditions

| Condition | FID↓ | CLIP↑ | SSIM↑ | PSNR↑ | LPIPS↓ |
|---|---|---|---|---|---|
| **High Temperature** | 115.33 | 0.35 | **0.47** | 15.52 | 0.64 |
| **Low Temperature** | 145.10 | 0.35 | 0.26 | 11.78 | 0.756 |
| **High Precipitation** | 170.80 | **0.40** | 0.36 | 11.92 | 0.72 |
| **Low Precipitation** | 85.62 | 0.33 | 0.43 | 15.10 | 0.66 |
| **High Radiation** | 107.34 | 0.36 | 0.46 | **15.83** | **0.64** |
| **Low Radiation** | 141.37 | 0.32 | 0.25 | 11.65 | 0.77 |

$\geq 25 \, \mathrm{W/m^2}$). This setup enables a focused assessment under rare climate scenarios. Results across standard metrics are shown in Table 2.

### 5.3. Multi-conditional Image Generation

Compared to the text-to-image generation models, multi-conditional generation achieves superior performance across all metrics, benefiting from added control signals that enhance its ability to generate more precise images while preserving both spatial structure and environmental factors. For evaluation in Tab. 3 we use the same test set as in the text-to-image generation task. As seen in Fig. 5, the generated images exhibit strong spatial alignment with the ground truth images, maintaining key geographical features while incorporating climate-specific changes. For instance, the shift in color grading from brown to green illustrates reasonable simulations of seasonal transitions, such as the onset of spring or the growth of vegetation after rainfall, which can also be inferred from increasing precipitation values in the images. The presence of snow in colder conditions reflects the model's responsiveness to low-temperature climate prompts, accurately simulating seasonal changes that would occur naturally in such environments.

Moreover, we can observe a noticeable shift in grasslands, transitioning from green to brown as temperature and solar radiation values increase. This shows a realistic response to environmental conditions such as drought or extreme heat. In certain land classes, such as barren and open shrublands, we do not observe significant changes, even with higher temperatures. These land types are relatively uniform throughout the year, which explains the minimal visual differences under varying climate conditions. The model captures this consistency by generating largely unchanged imagery, accurately reflecting their year-round stability, but also indicating a correlation between location, land cover, and the generated image.

For instance, when simulating snowy conditions in arid environments such as the United Arab Emirates, the generated images show only minimal or no adaptation to the extreme climate input.

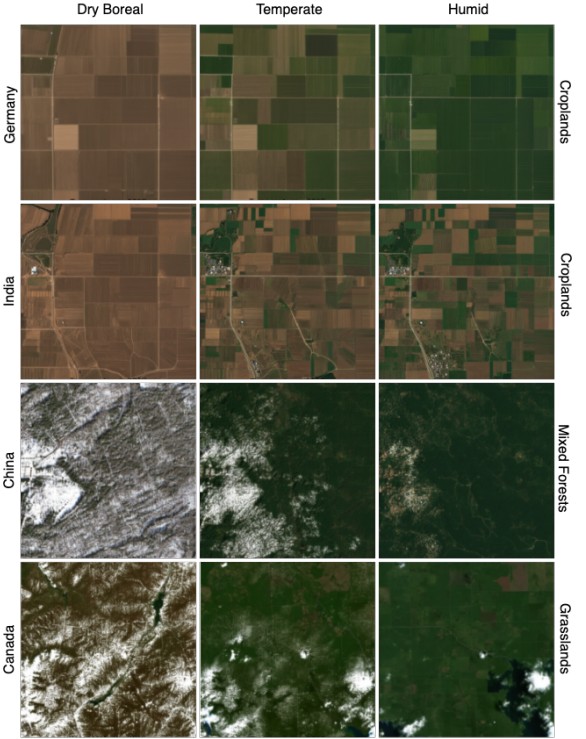

*Figure 4.* Satellite images generated by SD3-FT under extreme climate conditions for different regions. Rows correspond to geographical locations (Germany, India, China, and Canada), while columns represent three climate types: Dry Boreal, Temperate, and Humid. Land types (Croplands, Mixed Forests, Grasslands) are labeled on the right.

To complement the visual analysis, we conducted a quantitative stress test of the SD3-FT model on samples exhibiting extreme weather conditions. These were defined by thresholds: temperature ($\leq -10\,^{\circ}\mathrm{C}$ or $\geq 30\,^{\circ}\mathrm{C}$), precipitation ($\leq 0.01\,\mathrm{mm}$ or $\geq 10\,\mathrm{mm}$), and radiation ($\leq 5\,\mathrm{W/m^2}$ or

*Table 3.* Metrics for Stable Diffusion 3 ControlNet model

| Model | FID ↓ | CLIP ↑ | SSIM ↑ | PSNR ↑ | LPIPS ↓ |
|---|---|---|---|---|---|
| **SD3 ControlNet** | 48.20 | 0.32 | 0.40 | 13.63 | 0.59 |

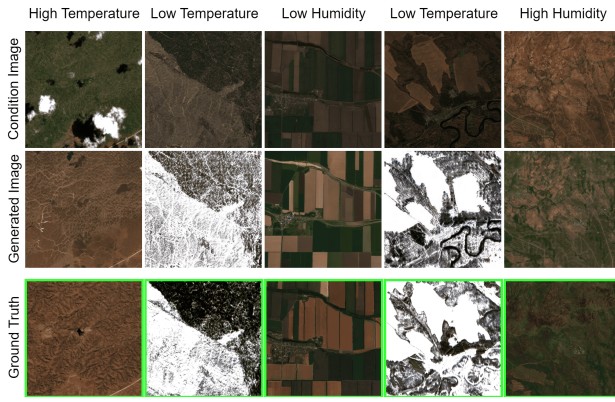

*Figure 5.* Comparison of Ground Truth, Generated Image, and Condition Image for multi-conditional image generation under seasonal changes. The top row displays condition images, while the labels above correspond to the ground truth images in the bottom row. All examples reflect typical seasonal variations (e.g., temperature and humidity) rather than extreme weather events. Color shifts (e.g., brown to green vegetation) and the presence or absence of snow illustrate the model's sensitivity to climate conditions, demonstrating its ability to generate seasonally consistent satellite imagery.

### 5.4. Robustness Test

#### 5.4.1. PERFORMANCE ACROSS LAND COVER TYPES

We further analyze the model's consistency across diverse land cover types to assess how surface characteristics influence generative performance. Table 4 presents a subset of results, highlighting both frequently occurring and challenging landcover classes. Grasslands and savannas - well-represented in the training data - achieve stable and high-fidelity generations, whereas visually complex or underrepresented classes such as wetlands and urban areas yield lower scores. The complete per-class evaluation is provided in Appendix Table 7.

*Table 4.* SD3_FT performance across selected land cover types. Dataset % indicates class prevalence in the test set (2017–2024).

| Land Cover | FID↓ | CLIP↑ | SSIM↑ | PSNR↑ | LPIPS↓ | Dataset % |
|---|---|---|---|---|---|---|
| Grasslands | 121.93 | 0.32 | 0.38 | 14.42 | 0.71 | 23.7% |
| Savannas | 116.90 | 0.35 | 0.38 | 13.84 | 0.71 | 12.0% |
| Permanent Wetlands | 263.36 | 0.51 | 0.39 | 13.73 | 0.73 | 0.8% |
| Urban/Built-Up | 284.65 | 0.70 | 0.06 | 11.17 | 0.81 | 0.7% |

#### 5.4.2. SPATIAL AND TEMPORAL ROBUSTNESS

Robustness in climate-aware satellite image generation can be viewed along two axes: spatial generalization producing realistic outputs at previously unseen locations and temporal generalization handling climate conditions from time periods outside the training range. Our test set design allows us to evaluate both. All 5,500 test locations are unique and globally sampled across the full time span from 2017 to 2024, while the training set is limited to data from 2017-2022. This ensures that test samples from 2023 and 2024 represent unseen combinations of both space and time.

As shown in Table 5, the model maintains consistent performance across all years, with no substantial degradation on 2023-2024 data.

*Table 5.* Robustness analysis for Controlnet

| Year | FID↓ | CLIP↑ | SSIM↑ | PSNR↑ | LPIPS↓ | Dataset % |
|---|---|---|---|---|---|---|
| **2017** | 102.14 | 0.36 | 0.40 | 13.33 | 0.61 | 8.48% |
| **2018** | 84.12 | 0.35 | 0.41 | 14.00 | 0.58 | 12.64% |
| **2019** | 89.52 | 0.36 | 0.41 | 13.90 | 0.58 | 12.38% |
| **2020** | 86.37 | 0.35 | 0.40 | 13.40 | 0.59 | 12.50% |
| **2021** | 83.30 | 0.33 | 0.40 | 13.73 | 0.58 | 12.14% |
| **2022** | 77.47 | 0.33 | 0.40 | 13.27 | 0.58 | 14.83% |
| **2023** | 93.21 | 0.35 | 0.43 | 14.09 | 0.59 | 9.78% |
| **2024** | 73.54 | 0.32 | 0.40 | 13.48 | 0.59 | 17.34% |

## 6. Discussion

In this study, we analyze the ability of diffusion models to integrate climate effects into satellite image generation. Through sensitivity experiments, we evaluate whether the models are able to capture meaningful climate-related transformations. The results show a consistent increase in simulated vegetation under warmer and more humid conditions, while cold and dry climates produce sparser vegetation or snow-covered landscapes. This trend aligns with known environmental patterns, where higher precipitation leads to denser vegetation, whereas arid and frigid conditions result in land cover dominated by barren surfaces or snow accumulation (Richardson et al., 2013). These observations indicate that diffusion models can learn the relationship between climate variables and environmental features and incorporate these patterns into satellite image synthesis. The sensitivity of the models varies depending on the land cover type. Evergreen zones, such as tropical rainforests, or urban areas, exhibit lower sensitivity to extreme climate prompts than regions with pronounced seasonal changes. This suggests that certain land cover types and locations inherently exhibit less variability in their visual characteristics, making climate-induced transformations less prominent. Extreme climate attack scenarios, such as generating snow-covered rainforests, may not represent realistic environmental conditions. We argue that correlations between locations or land cover type and typical climate conditions are learned during joint training. Therefore, a snowy rainforest, although both features (snow and rainforest) are individually present in the dataset, represents an out-of-distribution generation. Future work could further refine prompt masking strategies (e.g. partially omitting location or land cover specific informa-

tion) for increasing latent space interpolation capabilities.

The ablation studies on land cover types and spatiotemporal robustness support the expected behavior of the model and further validate its generalization capabilities. SD3-FT shows strong performance on well-represented land classes like grasslands and savannas, while lower scores are observed for visually complex or underrepresented categories such as wetlands and urban areas. This pattern aligns with findings from prior work on dataset imbalance in generative modeling (Ghosh et al., 2024; Qin et al., 2023) and remote sensing tasks (Leichtle et al., 2017; Xiao et al., 2024). The model also maintains stable performance across both spatially and temporally unseen samples, including new climate conditions from 2023 and 2024. This indicates that EcoMapper is applicable across varying locations, years, and land cover types - supporting that a consistent behavior can be expected in future scenarios.

The diffusion model's capability of mapping climate to satellite images is also reported in previous work, such as Leinonen et al. (2023) and Gao et al. (2023), who successfully integrated weather conditions into satellite imagery, although their focus was on nowcasting precipitation. A direct comparison is difficult, as to the best of our knowledge, no prior work has explicitly studied global climate-conditioned satellite image generation using diffusion models. Compared to cloud removal tasks, which typically report FID scores exceeding values of 120, our results suggest that generating climate-conditioned satellite images is a more learnable task, which can likewise be used to generate cloud-free images (Liu et al., 2024b). When compared to a broader range of general-purpose satellite image synthesis models, our FID scores are in line with models from other domains. In super-resolution tasks, authors have reported FID scores of 32 (Zheng et al., 2024), 29-40 (Wang & Sun, 2025), and about 16-25 from Khanna et al. (2024), Toker et al. (2024) and Yu et al. (2024b) in image generation tasks. Our metrics slightly exceed these values, suggesting that generating climate-conditioned visual effects on Earth's surface might be a moderately complex task, potentially requiring additional fine-tuning strategies, larger models or datasets.

Our experiments highlight the importance of pretraining on remote sensing data for climate-aware satellite image generation. Although DiffusionSat was not explicitly trained on climate data, it still outperformed the general-purpose SD3 model, suggesting that prior exposure to satellite imagery provides a significant advantage. Fine-tuning on climate metadata improved all models, reinforcing the benefits of incorporating climate information into satellite image generation. However, even after fine-tuning, DiffusionSat-FT maintained a slight edge over SD3-FT, likely due to its specialized satellite image generation capabilities. Despite this, the ability of SD3 to be trained at higher resolutions

(1024×1024) resulted in the best-performing model SD3-FT-HR, demonstrating that more details and finer textures were generated effectively. We demonstrated that multi-conditional generation using ControlNet preserves climate conditioning while introducing spatial guidance, making it suitable for generating synthetic satellite images for specific locations (e.g., areas of interest) under historical or future climate scenarios. This capability is particularly valuable for applications such as climate modeling, disaster response, and environmental monitoring, where reliable synthetic imagery can support long-term scenario analysis.

## 7. Conclusion

In this study, we introduced the comprehensive EcoMapper models and dataset linking climate data with corresponding Sentinel-2 satellite imagery and land cover types, enabling the generation of climate-conditioned satellite images. Our results demonstrate that diffusion models effectively map climate effects onto remote sensing imagery, with satellite-specific models outperforming general-purpose diffusion models. These findings highlight the potential of diffusion models as a valuable tool for generating past and future climate-conditioned satellite images, supporting applications in agriculture, forestry, climate change analysis and many other domains. Future research could explore higher time-resolution climate inputs to enhance the accuracy of climate-Earth surface interactions. Extending this approach to multispectral image generation would increase its applicability to environmental monitoring, where vegetation indices rely on both visible and non-visible spectral bands. Linking diffusion-based image generation with climate projection models could enable seasonal climate simulations, providing insights into potential land surface changes under different climate scenarios. These advancements would further solidify generative models as a powerful tool for climate-aware Earth observation.

## 8. Code and Data Availability

The code is available under: https://github.com/maltevb/ecomapper.
The dataset is available under: doi:10.14459/2025mp1767651

## Acknowledgement

We thank the European Space Agency Network of Resources and Horsch Maschinen GmbH for support to acquire the satellite imagery dataset.

## Impact Statement

EcoMapper introduces a generative framework for simulating satellite imagery conditioned on climate variables, with the goal of modeling how environmental landscapes respond to weather and long-term climate change. This enables new opportunities for climate change impact visualization, scenario exploration, and enhancing downstream models that integrate satellite and climate data-such as crop yield forecasting, land use monitoring, or image gap-filling in cloudy regions. We acknowledge that synthetic imagery carries risks of misinterpretation if used without proper context or uncertainty modeling. This is particularly relevant in high-stakes settings like natural disaster assessment or policy-making, where unreliable outputs could cause harm. We encourage the responsible use of EcoMapper as a simulation tool to complement - rather than replace - physical Earth observation in real-world decision-making.

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

# A. Appendix

## A.1. Data

The Ecomapper dataset consists of over 2.9 million satellite images with climate metadata. It includes RGB imagery and corresponding metadata from the Sentinel satellite missions, covering various land cover types and temporal data points. The training set contains 98,930 locations, each spanning 24 months of data, while the test set includes 5,494 locations, each covering 96 months. Tab. 6 has details about the structure.

*Table 6.* Yearly data for batches from 2017 to 2022, showing the number of locations in each batch across three time groups. Each time group consists of three batches, with two large batches of 14,000 locations and one smaller batch of 5,000 locations, totaling 33,000 locations per group. Each batch sampled randomly according to the earth's distribution of landcover types. The test batch, with 5,500 locations, is included for all eight years, while the other batches cover two years each. The total image count across all batches is 2,904,000. Note: There are some missing locations due to the fitting of the earth's land cover distribution.

| Batch/Year | 2017-2018 | 2019-2020 | 2021-2022 | 2023-2024 | Total Image |
|---|---|---|---|---|---|
| **Batch-1, 2, 3** | 33,000 | 0 | 0 | 0 | 792,000 |
| **Batch-4, 5, 6** | 0 | 33,000 | 0 | 0 | 792,000 |
| **Batch-7, 8, 9** | 0 | 0 | 33,000 | 0 | 792,000 |
| **Test Batch** | 5,500 | 5,500 | 5,500 | 5,500 | 528,000 |
| **Total Image** | 924,000 | 924,000 | 924,000 | 132,000 | 2,904,000 |

For each temporal data point, the metadata also provides weather data, including temperature, solar radiation, and precipitation. The satellite imagery originates from the Copernicus Sentinel-2 mission, supported by the European Space Agency (ESA) NoR program. The climate data, consist of average monthly temperature, solar radiation, and total precipitation, gathered from NASA Power Api (NASA, 2025).

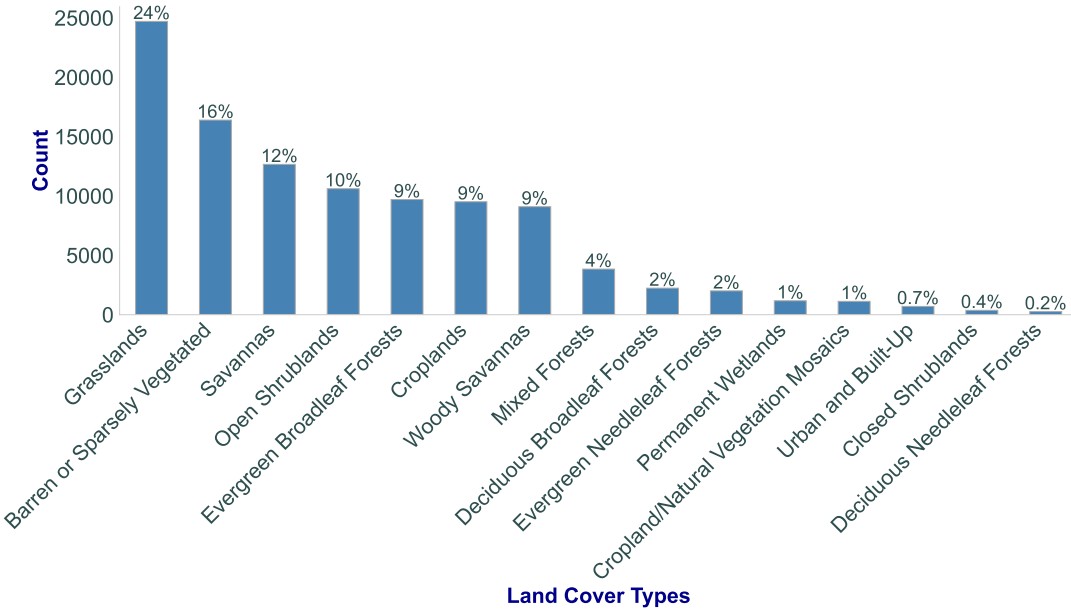

*Figure 6.* The distribution of land cover types, showing the total count for each category. The dataset includes 15 distinct land cover types, sampled using Google Earth Engine land cover maps from Phan et al. (2020).

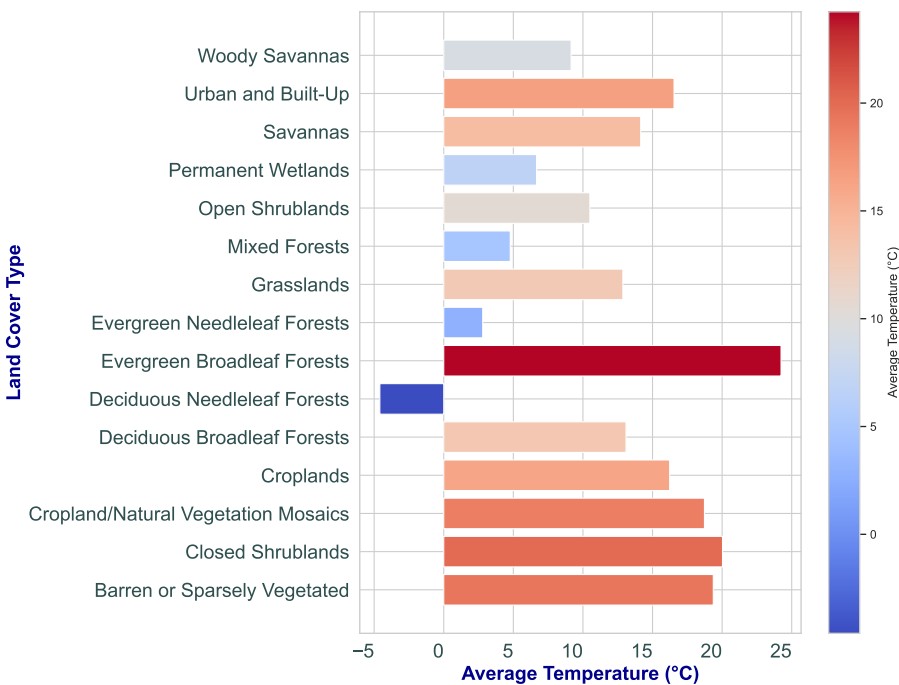

*Figure 7.* Average temperature for each land cover type. The temperature data was gathered from the NASA Power Api, representing monthly average temperatures across the sampled locations.

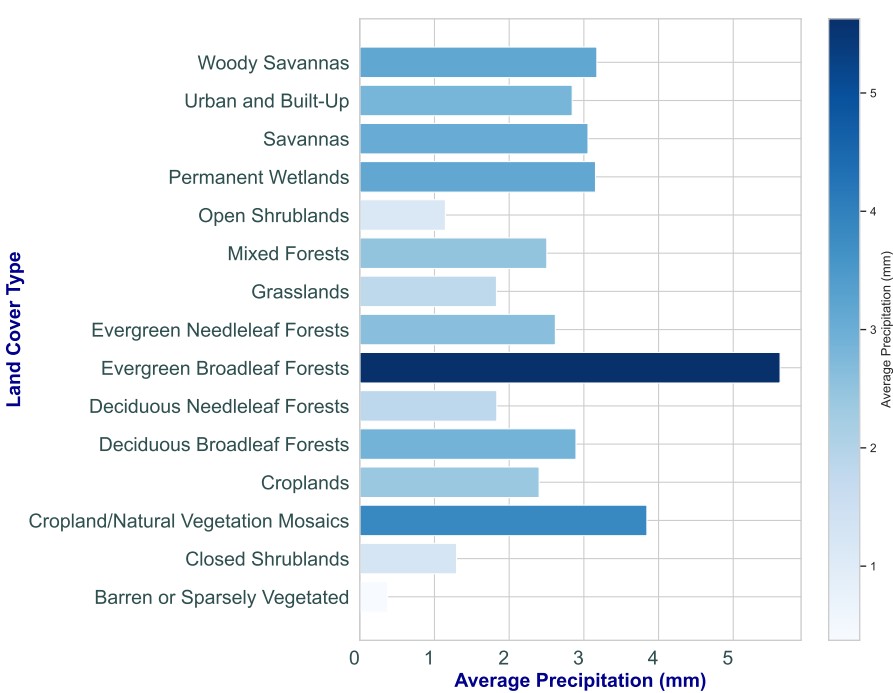

*Figure 8.* Average precipitation for each land cover type. The precipitation data was gathered from the NASA Power Api, representing monthly average precipitation across the sampled locations.

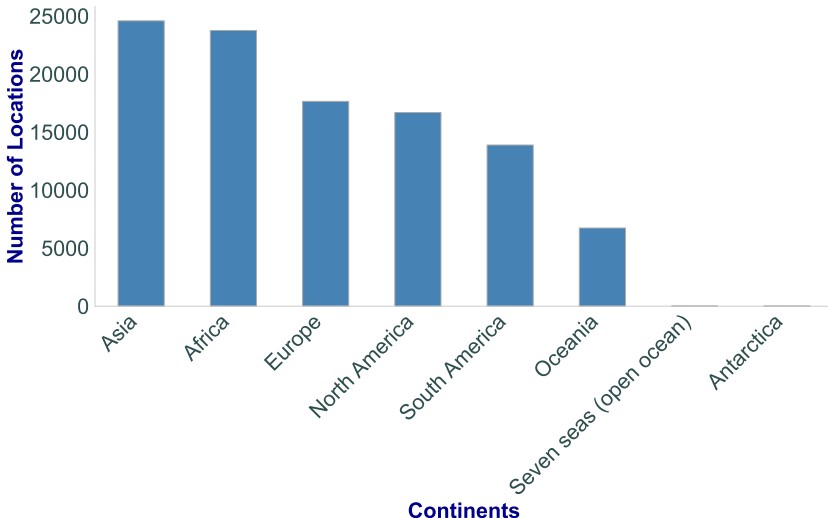

*Figure 9.* The distribution of locations across continents, illustrating the total count of locations within each continent. Water bodies and Antarctica were excluded from the sampling area.

### A.2. Evaluation Metrics

To comprehensively assess the quality of the generated satellite images, we employ the following evaluation metrics:

1. Learned Perceptual Image Patch Similarity (LPIPS) (Zhang et al., 2018): Measures perceptual similarity between images using deep feature embeddings. Lower values indicate higher similarity, aligning closely with human perception.
2. Fréchet Inception Distance (FID) (Heusel et al., 2017): Evaluates realism by comparing feature distributions of real and synthetic images using deep network embeddings. Lower values indicate more realistic images.
3. Structural Similarity Index (SSIM) (Wang et al., 2004): Assesses structural similarity between images based on luminance, contrast, and texture. Higher values (closer to 1) indicate greater similarity.
4. CLIP Score (Radford et al., 2021): Evaluates how well generated images align with textual descriptions using CLIP embeddings. Higher values indicate stronger alignment with the provided text prompt.
5. Peak Signal-to-Noise Ratio (PSNR) (Wang et al., 2004): Measures pixel-wise reconstruction accuracy using mean squared error. Higher values indicate lower distortion and better image quality.

### A.3. Experiments

In this section, we outline the experimental setup for fine-tuning three distinct models: DiffusionSat, SD3, and an enhanced variant of DiffusionSat incorporating additional climate metadata. Each model was fine-tuned to generate *climate-aware satellite imagery*, conditioned on both geospatial and climate data. By leveraging these models, we aim to produce more accurate and contextually relevant satellite images, integrating climate-specific information to improve the representation of dynamic environmental conditions.

#### A.3.1. FINE-TUNING DIFFUSIONSAT (TEXT-TO-IMAGE GENERATION)

For the DiffusionSat experiments, we started with the pre-trained checkpoint of the DiffusionSat_512 model, which had been trained for 150,000 iterations. This checkpoint was originally fine-tuned on a dataset of 512 images. To enhance the model's capability to capture climate-related information, we added 3 additional metadata features temperature, solar radiation, and precipitation bringing the total number of metadata features to 10. These additional layers were incorporated into the SatUNet architecture, which serves as the backbone of DiffusionSat.

The architecture was initialized with 10 metadata layers, with the weights from the pre-trained DiffusionSat_512 checkpoint

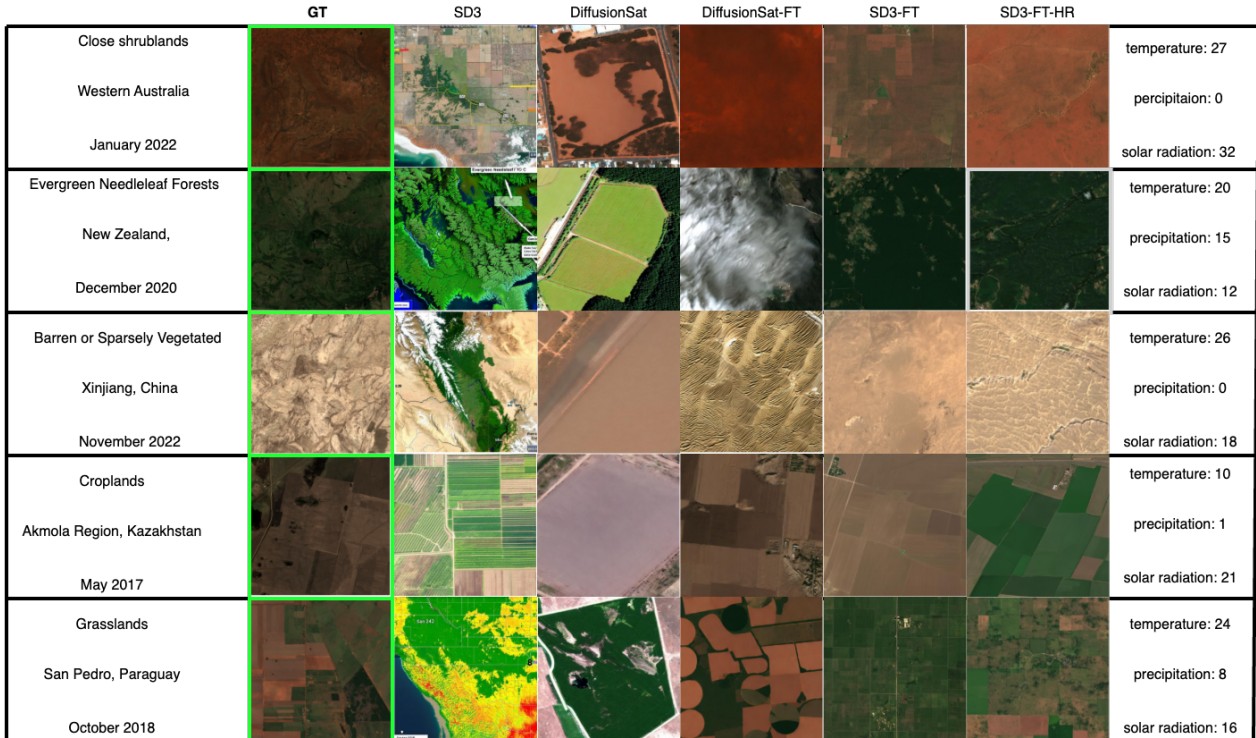

*Figure 10.* Comparison of generated satellite images from various models, including base models and fine-tuned models. The columns show images from the following models: Ground Truth (GT), SD3, DiffusionSat, DiffusionSat-FT, SD3-FT, and SD3-FT-HR, where FT denotes fine-tuned and HR high-resolution models. Each row represents a different geographical region, with associated metadata provided for the Ground Truth images, including temperature, precipitation, and solar radiation. The rows correspond to the following regions: (1) Close Shrublands in Western Australia (January 2022), (2) Evergreen Needleleaf Forests in New Zealand (December 2020), (3) Barren or Sparsely Vegetated in Xinjiang, China (November 2022), (4) Croplands in Akmola Region, Kazakhstan (May 2017), and (5) Grasslands in San Pedro, Paraguay (October 2018). The metadata on the right of each image includes the temperature (°C), precipitation (mm), and solar radiation (W/m²) for each location. This figure highlights the performance improvements achieved by fine-tuned models compared to their base counterparts.

loaded into the model. The three newly added metadata layers were initialized randomly following a normal distribution $N(0, 1)$, while the rest of the layers retained their original weights from the pre-trained model. This allowed the model to retain the general learning from the pre-trained checkpoint while adapting to the new climate data.

We fine-tuned this modified all layers of the model i.e. 900 million parameters on our training set, which consists of approximately 2 million satellite images, for 2 epochs. During fine-tuning, the encoder, decoder, and the CLIP text encoder were kept fixed, and only the new metadata layers were updated.

### A.3.2. FINE-TUNING SD3 (TEXT-TO-IMAGE GENERATION)

For the SD3 model, we fine-tuned two versions: SD3-FT and SD3-FT-HR.

For the fine-tuning of the SD3-FT model, we applied LoRA (Low-Rank Adaptation) (Hu et al., 2022) by adding adapter layers to the top of the Transformer block, which introduced approximately 18 million trainable parameters. LoRA effectively reduces the number of trainable parameters while maintaining high model performance. Despite this reduction, we observed blurry output images after fine-tuning. To address this issue, we fine-tuned the last two layers of the model the projection layer and the normalization layer since these layers are crucial for output quality, particularly in the final stages of the image generation process. The projection layer is responsible for mapping internal representations to the output space, while the normalization layer ensures stable and high-quality output distributions. Fine-tuning these layers helped improve the sharpness of the generated images. The model was trained for 2 epochs on our training set.

In addition, we implemented a captioning strategy during training to introduce variability in the input captions. This strategy aimed to reduce the direct correlation between location, month, and climate data in the model's learning process. Typically, satellite images exhibit a strong correlation between temporal information (like month and year) and environmental factors (like temperature). For example, January in Russia is usually cold, and this information can be easily inferred from the month metadata. However, we wanted the model to learn to generate images based on climate data (such as temperature, precipitation, and solar radiation) independently of the month and location, allowing it to focus more on the environmental conditioning provided by the climate prompt.

Specifically, for each caption in the dataset, we randomly altered the inclusion of certain metadata components. With a 50% chance, we either removed the month or the location from the caption. More specifically, for the month and year:

- 25% of the time, we removed both the month and the year.

- 50% of the time, we included both the month and the year.

- 25% of the time, we included only the year.

For the location metadata:

- 25% of the time, we removed both the state and country.

- 50% of the time, we included both the state and country.

- 25% of the time, we included only the country.

This randomization was applied every time the caption was generated during training, ensuring a diverse set of training inputs and helping the model generalize better to varying levels of spatial and temporal metadata availability.

This approach was validated in the extreme case experiment 4, where we showed that by removing the month from the caption and changing only the climate data, the generated images exhibited significant differences. This experiment demonstrated that the model could still generate distinct images based on climate variables, even when the month metadata was removed, reinforcing the idea that the model was learning to use climate data effectively rather than relying solely on temporal cues.

For SD3-FT-HR, we used fixed caption i.e. with all the spatial and climate data, and we did not use LoRA, opting instead to fine-tune 70 % of the Transformer blocks. Specifically, we fine-tuned 1.5 billion parameters, which made training computationally expensive. Additionally, we reshaped the training images from 512 to 1024 resolution, as recommended in the original SD3 paper for achieving the highest-quality outputs. Fine-tuning was performed on 60 % of our training set, which corresponds to 1.2 million images. Training on 1.5 billion parameters resulted in a much slower training process, taking 5–6 times longer compared to the SD3-FT model with LoRA. Despite the longer training time, this approach allowed the model to benefit from higher-resolution images, critical for capturing finer details in satellite imagery.

### A.3.3. FINE-TUNING SD3 CONTROLNET

For the SD3 ControlNet model, we chose SD3-FT as the prior due to its faster training time and lower computational cost, which was particularly important given our limited resources. The SD3-FT model was fine-tuned using 512x512 resolution images, striking a balance between performance and computational efficiency. This model was trained with LoRA (Low-Rank Adaptation) to reduce the number of trainable parameters while preserving high performance. The key advantage of SD3-FT is its efficient use of resources, as it requires fewer parameters compared to higher-resolution models, such as SD3-FT-HR, making it a more practical choice for our task.

To fine-tune the SD3 ControlNet model, we transferred the weights from the first 12 layers of the SD3-FT model and the LoRA adapter into ControlNet. We then fine-tuned all layers of ControlNet, including the LoRA adapter, while freezing the transformer layers. This resulted in fine-tuning 1.1 billion trainable parameters. The reason for fine-tuning not only the LoRA weights but also the ControlNet layers is that fine-tuning just the LoRA layers would not have been sufficient to preserve the spatial structures in the generated images, as the LoRA adapter alone has only 18 million parameters. By fine-tuning additional ControlNet layers, we ensure that the model captures the spatial information necessary for generating accurate and high-quality satellite imagery, especially when incorporating climate-specific changes.

We trained the model for 2 epochs on a dataset of approximately 2 million images, which allowed the model to effectively learn climate-aware image generation within our computational constraints. Although SD3-FT-HR has been shown to produce higher-quality images, we opted for SD3-FT to optimize training speed and resource usage, as the 512x512 resolution model offered an effective trade-off between image quality and computational efficiency. The results from fine-tuning SD3-FT demonstrated good performance in generating spatially accurate, climate-aware images, validating this approach.

### A.3.4. ABLATION – PERFORMANCE ACROSS LAND COVER TYPES

We conduct an ablation study to evaluate the robustness of SD-FT and ControlNet's generation performance across various land cover types. Specifically, we assess how well the model generalizes when conditioned on different control maps representing diverse land cover categories.

To this end, we compute performance metrics on the full 2017–2024 test set and report averages per land cover class. As shown in Table 7 and 8, generation quality tends to correlate with the prevalence of each class in the training data. Frequent classes such as *Grasslands* and *Savannas* yield more stable results, while less-represented classes like *Wetlands* and *Urban/Built-Up* exhibit degraded performance. This suggests that data distribution plays a significant role in generation robustness and fidelity.

*Table 7.* SD3_FT performance across land cover types. Dataset % indicates class prevalence in the test set (2017–2024).

| Land Cover | FID↓ | CLIP↑ | SSIM↑ | PSNR↑ | LPIPS↓ | Dataset % |
|---|---|---|---|---|---|---|
| **Grasslands** | 121.93 | 0.326 | 0.381 | 14.42 | 0.715 | 23.7% |
| **Barren/Sparse Vegetation** | **103.57** | 0.355 | **0.514** | **15.62** | **0.572** | 16.1% |
| **Savannas** | 116.9 | 0.35 | 0.38 | 13.84 | 0.71 | 12% |
| **Open Shrublands** | 136.08 | 0.337 | 0.362 | 14.64 | 0.754 | 9.9% |
| **Evergreen Broadleaf Forests** | 164.17 | 0.343 | 0.425 | 12.97 | 0.724 | 9.9% |
| **Woody Savannas** | 136.65 | 0.353 | 0.334 | 13.16 | 0.713 | 9.1% |
| **Croplands** | 181.38 | 0.336 | 0.383 | 14.32 | 0.725 | 9.0% |
| **Mixed Forests** | 200.36 | 0.357 | 0.309 | 14.06 | 0.751 | 3.6% |
| **Dec. Broadleaf Forests** | 250.01 | 0.402 | 0.369 | 14.76 | 0.759 | 2.1% |
| **Evergreen Needleleaf Forests** | 235.00 | 0.441 | 0.291 | 13.14 | 0.757 | 2.0% |
| **Crop/Nat. Veg. Mosaics** | 252.47 | 0.377 | 0.316 | 13.89 | 0.726 | 1.1% |
| **Permanent Wetlands** | 263.36 | 0.516 | 0.395 | 13.73 | 0.739 | 0.8% |
| **Urban/Built-Up** | 284.65 | **0.701** | 0.064 | 11.17 | 0.811 | 0.7% |
| **Closed Shrublands** | 340.73 | 1.489 | 0.402 | 16.02 | 0.779 | 0.4% |
| **Dec. Needleleaf Forests** | 366.61 | 1.874 | 0.256 | 13.36 | 0.759 | 0.2% |

These results emphasize the importance of class distribution balance in training datasets for generative models. Future work may explore targeted augmentation or class-aware sampling strategies to improve performance on underrepresented land cover types.

### A.3.5. ABLATION – PROMPTING STRATEGIES FOR CLIMATE CONDITIONING

We evaluated multiple prompting strategies to determine their effect on generation performance when conditioning the model on climate information. In all configurations, the CLIP encoder input was kept constant using a short spatial prompt to maintain consistent image-text alignment. Variations were introduced only in the T5 encoder input, which incorporated climate-related information.

The following strategies were compared:

- **Numerical Climate Data + Short Prompt**: Combined a spatial prompt (e.g., region and time) with continuous

*Table 8.* ControlNet performance across land cover types. Dataset % indicates class prevalence in the test set (2017–2024).

| Land Cover | FID↓ | CLIP↑ | SSIM↑ | PSNR↑ | LPIPS↓ | Dataset % |
|---|---|---|---|---|---|---|
| **Grasslands** | **65.27** | 0.33 | 0.40 | 14.06 | 0.60 | 23.7% |
| **Barren/Sparse Vegetation** | 67.44 | 0.32 | **0.60** | **20.18** | **0.40** | 16.1% |
| **Savannas** | 118.73 | 0.33 | 0.37 | 10.73 | 0.65 | 12% |
| **Open Shrublands** | 88.25 | 0.34 | 0.41 | 15.95 | 0.58 | 9.9% |
| **Evergreen Broadleaf Forests** | 122.37 | 0.37 | 0.39 | 9.13 | 0.68 | 9.9% |
| **Woody Savannas** | 109.99 | 0.33 | 0.31 | 10.32 | 0.64 | 9.1% |
| **Croplands** | 100.48 | 0.34 | 0.29 | 12.91 | 0.60 | 9.0% |
| **Mixed Forests** | 148.46 | 0.36 | 0.29 | 11.11 | 0.65 | 3.6% |
| **Dec. Broadleaf Forests** | 172.39 | 0.42 | 0.33 | 11.81 | 0.69 | 2.1% |
| **Evg. Needleleaf Forests** | 169.07 | 0.44 | 0.27 | 10.88 | 0.68 | 2.0% |
| **Crop/Nat. Veg. Mosaics** | 214.63 | 0.42 | 0.35 | 12.16 | 0.62 | 1.1% |
| **Wetlands** | 239.32 | 0.50 | 0.38 | 8.61 | 0.67 | 0.8% |
| **Urban/Built-Up** | 214.36 | **0.66** | 0.29 | 12.01 | 0.55 | 0.7% |
| **Closed Shrublands** | 215.98 | 0.47 | 0.18 | 8.28 | 0.62 | 0.4% |
| **Dec. Needleleaf Forests** | 297.67 | 0.52 | 0.28 | 12.17 | 0.65 | 0.2% |

climate variables such as temperature, precipitation, and solar radiation. This configuration yielded the best overall performance.

- **Categorical Climate Data + Short Prompt**: Climate variables were discretized into interpretable labels (e.g., "hot," "moderate," "extreme precipitation") and embedded in the prompt alongside the spatial component.

- **Numerical Climate Data + Short Prompt with Dropout**: Introduced stochasticity by randomly omitting segments of spatial and temporal metadata during training, simulating missing or uncertain input conditions.

- **Categorical Climate Data Only**: Used only categorical climate descriptors without the spatial prompt, requiring the model to rely entirely on environmental cues.

Table 9 reports results for the first three strategies under comparable conditions. The fourth strategy was omitted from benchmarking due to lower stability and limited relevance to our baseline.

*Table 9.* Evaluation of prompting strategies.

| Strategy | FID↓ | IS↑ | CLIP↑ | LPIPS↓ | PSNR↑ | SSIM↑ |
|---|---|---|---|---|---|---|
| **1. Numerical + Short Prompt** | 68 | 4.7 | 0.33 | 0.66 | 13.11 | 0.35 |
| **2. Categorical + Short Prompt** | 72 | 4.8 | 0.33 | 0.69 | 11.90 | 0.35 |
| **3. Numerical + Dropout + Short Prompt** | 94 | 4.5 | 0.35 | 0.71 | 13.30 | 0.35 |

These results highlight the advantage of preserving continuous climate signals in combination with spatial context. The dropout-based prompt introduces useful variability but at the cost of higher perceptual dissimilarity. A broader investigation of prompt design, including hybrid and dynamically adapted strategies, remains a promising direction for future work.

### A.3.6. RELATION TO SATCLIP

We acknowledge the relevance of SatCLIP (Klemmer et al., 2024) to our work and include a brief discussion here. SatCLIP is a contrastive learning framework for satellite imagery that aligns multi-spectral Sentinel-2 images with textual descriptions.

While its objectives are related to our use of CLIP-like models for image-text alignment, its architectural design and input requirements differ significantly.

Specifically, SatCLIP operates on 13-band Sentinel-2 imagery, whereas our dataset contains only RGB images. Adapting RGB data to match SatCLIP's input format would require artificial padding of missing channels, which can introduce noise and compromise the integrity of learned representations. To assess compatibility, we conducted a pilot evaluation using SatCLIP embeddings on our RGB-based generations and ground truth imagery.

*Table 10.* SatCLIP similarity scores for different models evaluated on RGB imagery. Lower (negative) scores indicate weaker alignment under SatCLIP's multi-spectral embedding space.

| Model | Avg. SatCLIP Score |
|---|---|
| **SD3_FT_HR** | -0.0148 |
| **SD3_FT** | -0.0105 |
| **Diffsat_FT** | -0.0285 |
| **Ground Truth** | -0.0171 |

The results confirm a significant mismatch: even ground truth RGB images yield negative SatCLIP scores. This outcome reflects the model's strong dependence on spectral features unavailable in our dataset, making it unsuitable as a reliable evaluation metric for RGB-only generative models. For this reason, we did not include SatCLIP as a core benchmark in our main analysis, though we appreciate the reviewer's suggestion and include it here for completeness.

