# OpenReview forum: "EcoMapper: Generative Modeling for Climate-Aware Satellite Imagery"
_ICML.cc/2025/Conference — ICML 2025 poster_

### Official Review · Reviewer_oS3e · 2025-03-04

**Overall Recommendation:** 4

**Summary:**

This paper introduces two generative models for the generation of controllable satellite images. The models, based upon Stable Diffusion 3, enable the controlled generation of satellite images with several control types, including: Image-conditioned generation, spatiotemporal conditioning (location, date, land cover type, clouds), and climate-control (temperature, radiation, precipitation), and a novel combination of these factors. For training these models, the paper introduces a novel dataset of 2.7 million images based on Sentinel-2, spanning more than 100k geographical locations. The model is evaluated against several baselines, across relevant metrics for image generation quality and alignment. The contributions of this paper are said to advance climate-aware earth observation.

## update after rebuttal
I have read the rebuttal and other reviews. The authors have adequately addressed my concerns, and have provided a solid rebuttal overall. I am more positive about this paper and have increased my rating.

**Claims And Evidence:**

The paper makes two main claims. First that they introduced a dataset for the training of satellite image generation models. This claim is supported by evidence, provided particularly on the supplementary material, albeit more analyses would be welcome. Then, the paper introduces two different models generative models to solve the aforementioned tasks. These are presented in the paper, and relatively well evaluated, but I am not convinced that they are particularly novel in the context of generative models.

**Essential References Not Discussed:**

I am missing a reference to SatCLIP, which I find to be closely related to the contents of this paper.

SatCLIP: Global, General-Purpose Location Embeddings with Satellite Imagery (Klemmer et al, 2023).

**Experimental Designs Or Analyses:**

I checked the experimental analyses in the paper. I found them to be sound in the context of the problem. The method is evaluated against several relevant baselines, across several metrics. Note that the metrics used in this paper are common for generative models (FID, PSNR, LPIPS, SSIM) and for text-conditioned models (CLIP). However, none of these metrics are particularly well-suited for the task this paper is trying to solve. I would suggest using SatCLIP as an additional metric for enhancing the quantitative results.

Further, the paper provides extensive qualitative results across different conditionings, showing the capabilities of the model. In any case, it would be interesting to do some sensitivity analysis, to study how much the images change in the dataset when the climate or temporal variables change, and compare with how much they change in the images generated by the model.

**Methods And Evaluation Criteria:**

The methods and evaluation criteria make sense int he context of the problem, I have no signifcant concerns about this (see my comments in Experimental Designs Or Analyses for a more thorough analysis of the evaluation part of this section).

Nevertheless, I remain unconvinced about the novelty of the model introduced by this paper. The conditioning mechanisms used in this paper are well studied in the literature, and using them for a new task does not constitute a novel technical contribution, in my opinion.

**Other Comments Or Suggestions:**

- I suggest the authors expand this dataset with more human-inhabited areas,

**Other Strengths And Weaknesses:**

- The paper is well structured, well written and the quality of the figures is generally high.
- Analysis of related work is sound.
- My main concern with this paper is the lack of technical novelty. While the dataset may prove valuable, the methods introduced in this paper are hardly novel, and therefore the impact of this paper may be limited to using the dataset or the generative models as tools. This limits the scope of the paper quite strongly, in my opinion. Nevertheless, this is an applications-driven paper and, as such, I believe it could have some impact.

**Questions For Authors:**

- What is the influence of scale and resolution in the model outputs when using image conditioning? Does it affect the outputs strongly?

**Relation To Broader Scientific Literature:**

To my eyes, this paper introduces a dataset and generative models for tasks that are somewhat novel, which are the climate-controllable generation of satellite imagery. This was underexplored in the literature.

**Theoretical Claims:**

There are no significant theoretical claims in this paper, it is very application-driven.

---

> ### Author Rebuttal · Authors · 2025-04-01
>
> **Reviewer comment:**
> I am missing a reference to SatCLIP, which I find to be closely related to the contents of this paper.
>
> **Response:**
>
> Thank you for this helpful suggestion. We agree that SatCLIP (Klemmer et al., 2023) is highly relevant and will include a citation and brief discussion in the revised version of the paper. We considered using SatCLIP in our evaluation. However, the model is designed for multi-spectral Sentinel-2 inputs (13 channels), while our dataset consists of standard RGB images (3 channels). To make our images compatible, we would need to artificially pad the missing bands, which introduces noise and may result in embeddings that do not reflect SatCLIP’s original intent. We nonetheless tested SatCLIP for comparison, and the results support our concerns. As shown below, even the ground truth RGB images receive low (negative) SatCLIP scores, suggesting a mismatch between the model's input expectations and our dataset.
>
> **Table 7: SatCLIP Scores on RGB Imagery**
> | Model       | Avg. SatCLIP Score |
> |-|-|
> | SD3_FT_HR   | -0.0148            |
> | **SD3_FT**  | **-0.0105**        |
> | Diffsat_FT  | -0.0285            |
> | Ground Truth| -0.0171            |
> These results suggest that SatCLIP is not well-suited for RGB-only satellite imagery, which is why we chose not to use it as a core evaluation metric. But we are happy to receive the reviewers feedback on this.
>
> **Reviewer comment:**
>
> I remain unconvinced about the novelty of the model introduced by this paper. The conditioning mechanisms used in this paper are well studied in the literature, and using them for a new task does not constitute a novel technical contribution, in my opinion.
>
> **Response**
>
> Thanks for the fair comment. As an application-focused paper, our aim is to demonstrate the feasibility of climate-aware satellite image synthesis using established models. Our contribution lies in adapting them for multi-conditional generation with structured climate prompts and continuous variables. We also added more discussion about the prompting strategy as outlined in the reply to reviewer 54H1, as we think this is a valuable contribution to the development of future climate- aware generative models for EO data.
>
> **Reviewer comment:**
>
> I am puzzled by the lack of urban and rural but inhabited lands. This somewhat limits the scope of the dataset, in my opinion.
>
> **Response:**
>
> Thank you for raising this point. Our dataset was constructed through globally uniform random sampling and reflects the class distribution of the underlying land cover map, which defines only 16 categories—of which “Urban and Built-Up” is the only inhabited land type explicitly represented. We did not manually adjust class frequencies, and the share of urban areas (~0.7%) aligns with the global land surface covered by this class. While inhabited areas are important, our primary focus is on landscapes where climate conditions are a key driver of visual and environmental variation. In urban settings, land cover changes are often dominated by human interventions (e.g., new buildings, roads), which are less predictable from climate variables alone. For this reason, we consider urban and densely inhabited areas to be of secondary relevance in the current version of the dataset.
>
> **Reviewer question:**
> What is the influence of scale and resolution in the model outputs when using image conditioning? Does it affect the outputs strongly?
>
> **Response:**
>
> Thank you for this technical question. Our ControlNet-based model operates at a fixed resolution of 512×512, which is a downscaled version of SD3’s native 1024×1024. During fine-tuning, we adjusted only the last layers, allowing the model to adapt its pre-trained high-resolution features to the lower resolution. In practice, using higher-resolution conditioning inputs could improve spatial detail and texture fidelity. However, since the model is trained entirely at 512×512, feeding 1024×1024 inputs without re-training could introduce inconsistencies or degrade visual quality due to the resolution mismatch. We agree that training at higher resolutions would likely enhance fine-grained spatial accuracy and believe it is a promising topic for future research activities.

---

> > ### Comment · Reviewer_oS3e · 2025-04-08
> >
> > I have read the rebuttal and other reviews. The authors have adequately addressed my concerns, and have provided a solid rebuttal overall. I am more positive about this paper and will increase my rating.

---

> > > ### Author Response · Authors · 2025-04-08
> > >
> > > We are pleased that we were able to address the concerns, thank you for the kind feedback and we are happily awaiting the rating upgrade.

---

### Official Review · Reviewer_vwzA · 2025-03-13

**Overall Recommendation:** 3

**Summary:**

This paper extends previous work on satellite image generation by introducing a larger dataset and considering two generation scenarios - text2img and ControlNet. Quantitative comparison shows superior performance only in FID but not other metrics (i.e., CLIP, SSIM, PSNR, LPIPS).

**Claims And Evidence:**

This paper presents EcoMapper, a large-scale satellite imagery dataset integrated with climate data. While the dataset's size is notable, the paper's primary contribution and the novelty of its proposed prompting setups remain unclear. The prompting methods, as presented, appear incremental. Furthermore, while potential applications like forecasting and scenario analysis are mentioned, the paper lacks concrete evidence demonstrating the utility of the generated imagery for these downstream tasks. Demonstrating the practical benefits of EcoMapper's generative capabilities would significantly enhance the paper's impact.

**Essential References Not Discussed:**

This work can be helpful for cloud removal in satellite imagery, and the reviewer can consider adding more references in this domain, such as SEN12MS-CR-TS (TGRS'22), AllClear (NeurIPS'24), or DiffCR (TGRS'24).

It would also be helpful if the authors discuss concurrent work such as CRS-Diff (TGRS'24) and MetaEarth (TGRS'24).

**Experimental Designs Or Analyses:**

One way to "convince" the reader of the contribution of this work in forecasting is predicting hold-out images or future images. Say the model had stopped training for one month, and then I would wonder how good or bad this model is in forecasting future imageries, conditioning on future climate data (which is more mature and well-studied). In addition to this, a detailed analysis upon why the model performs good or bad can be very helpful for future work on satellite imgery generation. Stratified analysis such as under what conditions the model performs worse (e.g. extreme hot weather, or snowy scene) can also provide more insights to this community. I am very willing to raise my score is this experiment is provided.

**Methods And Evaluation Criteria:**

Since it is a generative modeling task, the presented evaluation is standard.

**Other Comments Or Suggestions:**

NA

**Other Strengths And Weaknesses:**

NA

**Questions For Authors:**

NA

**Relation To Broader Scientific Literature:**

NA

**Theoretical Claims:**

NA

---

> ### Author Rebuttal · Authors · 2025-04-01
>
> Thank you for your detailed and constructive feedback. In the following, we address the key points you raised.
>
> **Reviewer comment:**
>
> While the dataset's size is notable, the paper's primary contribution and the novelty of its proposed prompting setups remain unclear.
>
> **Response:**
> The prompting strategy plays a key role in our approach. We tested several strategies for conditioning the model on climate and spatial context. These experiments are detailed in our response to Reviewer 54H1, where we present a breakdown of different prompt setups and their impact on generation quality. We kindly refer you to that section for a more in-depth explanation.
>
> **Reviewer comment:**
>
> One way to demonstrate forecasting value would be to evaluate the model on hold-out or future satellite images conditioned on climate data.
>
> **Response:**
>
> We conducted additional experiments to address this point. We evaluated all models on satellite images from 2023 and 2024, using the same 5,500 test locations as in the original setup. These years were not part of the training period (2017–2022), establishing a strict temporal split. The results (see below) show stable and consistent performance as in the other testings before, indicating that our models generalize well to unseen time periods. We will include this experiment, along with results for the single-image generation setup, in the final version of the paper. The 2023–2024 test data will be added to the dataset.
>
> **ControlNet – Year-wise Testset Results**
>
> |Year|CLIP↑|SSIM↑|PSNR↑|LPIPS↓|FID↓|Dataset share %|
> |-|-|-|-|-|-|-|
> |2017|0.36|0.40|13.33|0.61|102.14|8.5%|
> |2018|0.35|0.41|14.00|0.58|84.12|12.6%|
> |2019|0.36|0.41|13.90|0.58|89.52|12.4%|
> |2020|0.35|0.40|13.40|0.59|86.37|12.5%|
> |2021|0.33|0.40|13.73|0.58|83.30|12.1%|
> |2022|0.33|0.40|13.27|0.58|77.47|14.8%|
> |2023|0.35|0.43|14.09|0.59|93.21|9.8%|
> |2024|0.32|0.40|13.48|0.59|73.54|17.3%|
>
> We want to briefly clarify the use of the Fréchet Inception Distance (FID). FID is known to be less reliable on small test sets, as it estimates distribution similarity based on feature mean and covariance from a pre-trained Inception network. Limited sample size can introduce noise and variance in these estimates. We interpret FID cautiously and emphasize the importance of using it alongside other metrics to ensure a more robust evaluation.
>
> **Reviewer comment:**
>
> A detailed analysis of when and why the model performs well or poorly would be helpful. Stratified evaluation (e.g., under extreme weather conditions) could offer valuable insights. I am very willing to raise my score is this experiment is provided.
>
> **Response:**
>
> Thank you for this important suggestion. To address this, we performed additional stratified experiments focusing on extreme climate conditions, evaluating model performance across scenarios of high/low temperature, precipitation, and radiation. These weather extremes are closely tied to challenging land cover types (e.g., snowy forests, cloud-covered areas), which tend to be underrepresented in the training set and inherently harder to reconstruct. In contrast, warmer and drier regions — which are more common in the data — yield more accurate generations. The results below illustrate that low temperatures, high precipitation, and low radiation correspond to significantly lower visual fidelity and structural alignment, while high radiation and dry conditions lead to much better performance
>
> **Table 10: SD3_FT Performance Under Extreme Climate Conditions**
>
> |Condition|FID↓|SSIM↑|PSNR↑|LPIPS↓|Inception Score↑|CLIP Score↑|
> |-|-|-|-|-|-|-|
> |High Temperature|115.33|0.475|15.52|0.640|3.790|0.353|
> |Low Temperature|145.11|0.260|11.79|0.756|3.745|0.355|
> |High Precipitation|170.81|0.365|11.93|0.725|3.261|0.403|
> |Low Precipitation|85.63|0.430|15.10|0.665|4.185|0.339|
> |High Radiation|107.35|0.469|15.83|0.646|3.909|0.362|
> |Low Radiation|141.37|0.252|11.65|0.770|3.941|0.327|
>
> We also provide a detailed per–land cover class analysis in our response to Reviewer WAR9 (ControlNet)
>
> **Reviewer comment:**
> The reviewer suggests citing additional related work, including SEN12MS-CR-TS, AllClear, DiffCR, CRS-Diff, and MetaEarth.
>
> **Response:**
> We acknowledge the growing number of generative models and datasets in EO and will take up these works in the discussion section of our paper and include references in the revised version. SEN12MS-CR-TS and AllClear focus on cloud removal using multi-modal Sentinel-1/2 data but are limited either temporally (e.g., only 2022) or regionally. DiffCR uses diffusion for cloud removal but lacks temporal and climate conditioning. CRS-Diff introduces multi-modal control but operates at different resolutions and without climate or multi-temporal information. MetaEarth enables large-scale image generation but relies solely on image and geographic metadata. In contrast, our work uniquely combines image and continuous climate conditioning for globally distributed, climate-aware satellite image synthesis.

---

> > ### Comment · Reviewer_vwzA · 2025-04-08
> >
> > The authors have addressed my concerns with the additional experiments on temporally held-out experiments and stratification outcomes. I will raise my rating.

---

> > > ### Author Response · Authors · 2025-04-08
> > >
> > > Thank you for your kind feedback and thank you for the rating upgrade.

---

### Official Review · Reviewer_54H1 · 2025-03-14

**Overall Recommendation:** 1

**Summary:**

This paper introduces EcoMapper, which combines climate data with satellite imagery based on Sentinel-2 images. Satellite images often face observational challenges, such as areas affected by cloud cover and inherent resolution limitations that can hinder accurate analysis. To overcome these issues, the authors propose a text-prompt-based generative model that leverages both spatial and climate information to synthesize realistic satellite images. This approach promises to enhance the utility of satellite data in applications like environmental monitoring, climate change prediction, and agricultural planning.

**Claims And Evidence:**

The authors claim that satellite images can be generated using text prompts containing spatial and climate information, and that the control image plays a crucial role in preserving the spatial structure. To support this claim, it would be beneficial to include experiments that analyze the generated images based on control images taken from different timelines of the same region.

**Essential References Not Discussed:**

There do not appear to be any essential references missing.

**Experimental Designs Or Analyses:**

The number of experiments seems insufficient to fully validate the model's overall generalization capabilities. In particular, there is a lack of analysis for complex regions, such as urban areas, where the data and spatial structures are more challenging.

**Methods And Evaluation Criteria:**

The authors conducted experiments using a sufficiently large Sentinel-2 image dataset and evaluated the quality of the generated images based on various quantitative metrics such as FID, PSNR, and SSIM. These performance evaluation methods are appropriate. However, given the unique challenges of the satellite imagery domain, it would be valuable to include assessments that measure how closely the generated images match real-world conditions.

**Other Comments Or Suggestions:**

No additional comments.

**Other Strengths And Weaknesses:**

Strength:
The paper introduces a deep learning model for generating realistic satellite images, which has potential for applications of satellite data.

Weaknesses:
1. Limited ablation studies on control images: The paper does not provide detailed ablation studies on the selection and impact of different control images. This analysis is essential to determine the robustness of the approach.
2. Text prompt methodology: The authors incorporate text prompts to condition the generative process, however, there is insufficient discussion on how to concretely define effective text prompts, and the experiments evaluating which types of text prompts yield the best performance are lacking.
3. Overall structure and generalizability: The overall structure does not appear particularly novel, and there are concerns regarding the generalizability of the approach

**Questions For Authors:**

Questions are embedded in the above section.

**Relation To Broader Scientific Literature:**

This paper presents a new text prompt-based approach capable of reflecting various environmental conditions. The authors propose a method to generate conditioned satellite images by simultaneously incorporating spatial information and climate data.

**Theoretical Claims:**

In this paper, no new theoretical proofs or claims are directly addressed. Instead, the model is built upon existing research, such as  Stable Diffusion, and its performance is verified through experiments.

---

> ### Author Rebuttal · Authors · 2025-04-01
>
> We thank the reviewer for the feedback. We would like to address the main critics:
>
> **Reviewer:**
> The paper does not provide detailed ablation studies on the selection and impact of different control images.
>
> **Response:**
> While it's not fully clear what is meant by "selection and impact" of control images, we interpret this as a question of robustness. To address this, we conducted further analysis of the ControlNet model's performance across land cover types (2017–2024 test set). Results show that classes with higher representation tend to generalize better, while less frequent classes are more challenging to generate consistently.
>
> **Controlnet landcover types**
> |Land Cover|CLIP↑|SSIM↑|PSNR↑|LPIPS↓|FID↓|Dataset %|
> |-|-|-|-|-|-|-|
> |Grasslands|0.33|0.40|14.06|0.60|65.27|23.7%|
> |Savannas|0.33|0.34|10.55|0.65|94.00|21.1%|
> |Barren/Sparse Veg.|0.32|0.60|20.18|0.40|67.44|16.1%|
> |Open Shrublands|0.34|0.41|15.95|0.58|88.25|9.9%|
> |Evg. Broadleaf Forests|0.37|0.39|9.13|0.68|122.37|9.9%|
> |Woody Savannas|0.33|0.36|10.32|0.64|109.39|9.1%|
> |Croplands|0.34|0.29|12.91|0.60|100.48|9.0%|
> |Mixed Forests|0.36|0.29|11.11|0.65|148.46|3.6%|
> |Dec. Broadleaf Forests|0.42|0.33|11.81|0.69|172.39|2.1%|
> |Evg. Needleleaf Forests|0.44|0.27|10.88|0.68|169.07|2.0%|
> |Crop/Nat. Veg. Mosaics|0.42|0.35|12.16|0.62|214.63|1.1%|
> |Wetlands|0.50|0.38|8.61|0.67|239.32|0.8%|
> |Urban/Built-Up|0.66|0.29|12.01|0.55|214.36|0.7%|
> |Closed Shrublands|0.47|0.18|8.28|0.62|215.98|0.4%|
> |Dec. Needleleaf Forests|0.52|0.28|12.17|0.65|297.67|0.2%|
>
>
> **Reviewer:**
> *It would be valuable to include assessments that measure how closely the generated images match real-world conditions.*
>
> **Response:**
>
> We use established metrics such as FID, SSIM, and LPIPS, which are widely applied in both general and remote sensing image generation tasks — including in DiffusionSat (Khanna et al., 2023), which serves as a foundation for our work. This helps ensure that both visual quality and geographic realism are captured. We acknowledge that downstream evaluation  would be a valuable addition. We want to highlight a more detailed answer at reviewer WAR9.
>
> **Reviewer:**
>
> No supplementary material was provided
>
> **Response**
>
> We would like to clarify that we included supplementary material starting on page 12 of the main submission.
>
> **Reviewer:**
>
> Text prompt methodology: The paper lacks sufficient discussion and comparison of different prompting strategies
>
> **Response:**
>
> We appreciate this valid point and extend our appendix to include details on the development and evaluation of our prompting strategies. Across all experiments, the CLIP encoder input was kept constant using a short spatial prompt to ensure consistent image-text alignment. Variations were introduced only in the T5 encoder input, which handled climate-related information. We evaluated the following strategies:
>
> - **Numerical Climate Data + Short Prompt** (**best performance**)
>   Combined a short spatial prompt with continuous climate variables (e.g., temperature, precipitation, solar radiation).
>
> - **Categorical Climate Data + Short Prompt**
>   Climate variables were discretized in the prompt into human-readable categories (e.g., “hot,” “moderate,” “extreme precipitation”).
>
> - **Numerical Climate Data + Short Prompt with dropout**
>   Introduced stochasticity by randomly dropping parts of the spatial and temporal metadata from the prompt.
>
> - **Categorical Climate Data Only**
>   Used only categorical climate descriptors without the spatial prompt, encouraging the model to rely entirely on environmental signals.
>
> Due to computational constraints, we leave a broader investigation of prompting strategies to future work and consider our results here as an initial guide for further exploration.
> Below we report results for first three prompting strategies under comparable conditions.
>
> | FID | Inception Score | CLIP | LPIPS | PSNR | SSIM | Prompting # |
> |-:|-:|-:|-:|-:|-:|-|
> |68|4.7|0.33|0.66|13.11|0.35|1|
> |72|4.8|0.33|0.69|11.90|0.35|2|
> |94|4.5|0.35|0.71|13.30|0.35|3|
>
> **Reviewer comment:**
>
> The number of experiments seems insufficient to fully validate generalization, especially for complex regions like urban areas. The overall structure does not appear novel
>
> **Response:**
> As an application-focused paper, our aim is to demonstrate the feasibility of climate-aware satellite image synthesis using established models. Our contribution lies in adapting them for multi-conditional generation with structured climate prompts and continuous variables. While our dataset includes some urban areas due to global sampling, our focus is on climate-sensitive landscapes where environmental change is linked to climate inputs. Urban dynamics, in contrast, are typically shaped by human decision-making and fall outside the primary scope of this work. For generalization, we refer to our detailed response to Reviewer WAR9, where we introduce a new temporal evaluation on 2023 and 2024 data and report consistent results.

---

### Official Review · Reviewer_WAR9 · 2025-03-14

**Overall Recommendation:** 4

**Summary:**

The paper introduces EcoMapper, a generative modeling framework designed to synthesize climate-aware satellite imagery. It provides two primary contributions:
- EcoMapper Dataset: A comprehensive dataset comprising 2.7 million Sentinel-2 RGB satellite images from 104,424 global locations, annotated with climate metadata (temperature, precipitation, solar radiation) and spanning 15 land cover types.
- Generative Modeling Approaches:
* Text-to-Image Generation: Uses fine-tuned Stable Diffusion 3 models conditioned on structured textual prompts (geographic location, date, and climate metadata) to generate realistic synthetic satellite images.
* Conditional Image Generation: Employs ControlNet, enabling guided generation of satellite, allowing realistic representation of climate-driven landscape evolution and seasonal variations.

Contributions include:
- Demonstrating the feasibility of generating realistic satellite imagery conditioned explicitly on climate information, validated through quantitative metrics (FID, CLIP, SSIM, PSNR, LPIPS) and qualitative examples.
- A sensitivity analysis illustrating the generative models' capability to reflect climate-induced visual changes across various land cover types and climate conditions.
- Showing improved generative performance through fine-tuning models with climate-specific metadata, especially highlighting the advantage of spatial conditioning via ControlNet for preserving geographical consistency.

## Update after rebuttal
Thank you for clearly addressing my main concerns. The authors have provided important clarifications that strengthen the paper:

- They've revised the introduction to better articulate the intended applications of their approach, including forecasting models, climate change visualization, and filling observational gaps.
- They've conducted a new temporal validation experiment using 2023-2024 data (beyond their original 2017-2022 training period), demonstrating good generalization across time periods, which is critical for climate-related applications.

These responses address my primary concerns about application clarity and temporal generalization. The EcoMapper dataset and framework represent a valuable contribution to climate-aware satellite imagery generation. Given these clarifications, I updated my recommendation to accept this paper.

**Claims And Evidence:**

The paper's main claims—that generative models can produce realistic climate-conditioned satellite imagery—are supported by clear qualitative and quantitative evidence. However, the claim that these synthetic images are directly useful for environmental monitoring, scenario planning, or policy-making lacks explicit evidence from real-world downstream tasks. The evaluation primarily focuses on visual realism rather than practical accuracy or usefulness, making this latter set of claims less convincingly supported.

**Essential References Not Discussed:**

None.

**Experimental Designs Or Analyses:**

Yes, I checked the experimental design for evaluating generated images. The experiments clearly compared different generative models using standard visual metrics, which makes sense for assessing image realism. However, the dataset uses random splits without considering time, making it unclear if the model can actually generalize to future climate conditions. This might overestimate how well the model truly performs in real-world extreme climate scenarios.

**Methods And Evaluation Criteria:**

The methods used (Stable Diffusion 3 and ControlNet for satellite image generation) and the dataset linking Sentinel-2 images with climate data are appropriate for generating realistic climate-conditioned images. The evaluation metrics (FID, CLIP, SSIM, PSNR, LPIPS) are suitable for measuring visual quality and realism. However, these metrics don't directly measure practical usefulness or accuracy for real-world applications like forecasting, mapping, scenario planning or environmental monitoring. Including additional tests or user evaluations that show practical utility would make the methods more convincing.

**Other Comments Or Suggestions:**

None

**Other Strengths And Weaknesses:**

Strengths:
- The paper combines satellite imagery and climate data using generative AI, which is original and timely.
- The new large-scale dataset (EcoMapper) is valuable for researchers working on related problems.
- The examples provided clearly illustrate the model’s ability to represent climate-driven changes visually, making it useful for education, communication, or scenario planning.

Weaknesses:
- The intended application (visualization, communication, public awareness) wasn't clearly stated upfront, causing some confusion.
- The evaluation doesn't clearly demonstrate real-world usefulness or accuracy beyond visual quality.
- The data splitting method doesn't explicitly consider time, limiting confidence in the results for forecasting/scenario planning scenarios.

**Questions For Authors:**

None

**Relation To Broader Scientific Literature:**

The paper builds on recent advancements in generative models (like Stable Diffusion) and applies them specifically to remote sensing data. Prior works have used similar models for satellite imagery tasks such as super-resolution, image-to-image translation, and cloud removal. This paper uniquely extends these ideas by conditioning the generative models explicitly on climate variables (temperature, precipitation, solar radiation), enabling visualization of climate-driven environmental changes. It also introduces the EcoMapper dataset, significantly expanding available satellite imagery datasets linked to climate metadata.

**Theoretical Claims:**

The paper does not include any theoretical proofs or formal mathematical claims.

---

> ### Author Rebuttal · Authors · 2025-04-01
>
> We thank the reviewer for their constructive feedback and thoughtful evaluation of our work. Below we respond to the key points raised.
>
> **Reviewer comment:**
> The paper lacks explicit evidence that the synthetic images are directly useful for downstream tasks like environmental monitoring or scenario planning.
>
> **Response:**
> We acknowledge this important point. Our current work focuses on demonstrating the feasibility of climate-aware generative modeling for satellite imagery — a foundational step for downstream applications such as forecasting, scenario simulation, or visualization. Operational deployment (e.g., crop yield prediction, land use modeling) requires additional task-specific pipelines, benchmark data sets and a full set of research experiments, which are beyond the current scope. We note that related work such as DiffusionSat similarly does also not evaluate downstream utility and probably would be an own full paper. However, our method extends prior work by enabling explicit control over continuous climate variables, making it well-suited for simulation-based use cases. Prior studies (e.g., Liu et al., 2024; Toker et al., 2024) have shown that synthetic remote sensing imagery can support applications such as cloud-gap filling, semantic segmentation, and training data augmentation. Strong performance across quantitative metrics (FID, SSIM, CLIP, SATCLIP) signals practical utility. Moreover, synthetic imagery becomes especially valuable where real observations do not exist — such as forecasting future landscapes under climate change. We consider this work a foundational step and are actively exploring downstream integrations in follow-up projects.
>
> **Reviewer comment:**
>
> The paper’s intended application (visualization, communication, public awareness) is not clearly stated in the introduction.
>
> **Response:**
>
> Thank you for this helpful suggestion. We revised the end of the introduction to clarify the core use cases of our approach. The updated text reads:
>
> > *“In this paper, we introduce a novel approach to generate satellite images conditioned on geographic-climate prompts using Stable Diffusion 3. Our method enables the simulation of how weather and climate affect Earth’s surface — generating synthetic images that can support forecasting models (e.g., crop yield prediction or land cover classification), visualize climate change models under various scenario assumptions, or fill observational gaps in regions affected by persistent cloud cover. The approach is globally applicable and generates realistic images with 10-meter spatial resolution across diverse vegetation types (e.g., cropland, broadleaf forests, savannas), using information about location, land cover type, and climate conditions.”*
>
> **Reviewer comment:**
>
> The dataset split does not consider time, limiting confidence in generalization to future conditions.
>
> **Response:**
>
> We agree that temporal independence is critical for evaluating generalization. To address this, we added a new experiment in which all models were evaluated on satellite imagery from **2023 and 2024** at the 5,500 test locations. These years were not included in the former data set (2017–2022), ensuring a strict temporal split. Results show that the model generalizes well across years (see table below). The performance in the new test data does not significant deviate from the other test settings. We will integrate this experiment into the final version of the paper, and release the test data as part of the dataset upon acceptance. Additionally, our globally sampled dataset reduces the risk of encountering entirely unseen climate scenarios, and our edge-case tests (see Fig. 5) further support the model’s robustness to outlier conditions (please see here reviewer vwzA.
>
> **ControlNet Year Metrics**
>
> | Year | CLIP ↑ | SSIM ↑ | PSNR ↑ | LPIPS ↓ | FID ↓ | Dataset share (%)|
> |-|-|-|-|-|-|-|
> | 2017 | 0.36   | 0.40   | 13.33  | 0.61   | 102.14 | 8.48%      |
> | 2018 | 0.35   | 0.41   | 14.00  | 0.58   | 84.12  | 12.64%     |
> | 2019 | 0.36   | 0.41   | 13.90  | 0.58   | 89.52  | 12.38%     |
> | 2020 | 0.35   | 0.40   | 13.40  | 0.59   | 86.37  | 12.50%     |
> | 2021 | 0.33   | 0.40   | 13.73  | 0.58   | 83.30  | 12.14%     |
> | 2022 | 0.33   | 0.40   | 13.27  | 0.58   | 77.47  | 14.83%     |
> | 2023 | 0.35   | 0.43   | 14.09  | 0.59   | 93.21  | 9.78%      |
> | 2024 | 0.32   | 0.40   | 13.48  | 0.59   | 73.54  | 17.34%     |
>
> **Reviewer comment:**
>
> No supplementary material provided.
> **Response:**
> We would like to clarify that supplementary material begins on page 12 of the main submission. It includes additional information about the dataset structure, evaluation metrics, fine-tuning procedures, and our prompting strategy. We hope this addresses the concern.

---

### Decision · Program_Chairs · 2025-05-01

**Decision:**

Accept (poster)

**Comment:**

This paper introduces a dataset of 2.7M set of Sentinel-2 RGB satellite images, and two Stable Diffusion-based models to text-to-image and ControlNet conditional generation, which are evaluated on the generation of satellite imagery for climate change visualisation applications.

Reviewers WAR9, oS3e and 54H1 found the application interesting and the dataset useful (confirmed by vwzA) as well as the paper well written and sound (oS3e).

Reviewer oS3e  had concern about technical novelty but agreed this was an application-driven paper. Reviewer WAR9 and vwzA found the application not clearly discussed and the real-world usefulness not demonstrated (clarified during rebuttal). Reviewer 54H1 found the ablations of control images insufficient and the text prompting not clearly evaluated (the authors provided ablation experiments to respond to these questions).

After rebuttals and discussion, all reviewers but one raised their score, and given the average score of 3 I recommend acceptance.